# SHAP Meets Tensor Networks:
# Provably Tractable Explanations with Parallelism

**Reda Marzouk**[*]
LIRMM, UMR 5506, University of Montpellier, CNRS
mohamed-reda.marzouk@umontpellier.fr

**Shahaf Bassan***
The Hebrew University of Jeursalem
shahaf.bassan@mail.huji.ac.il

**Guy Katz**
The Hebrew University of Jeursalem
g.katz@mail.huji.ac.il

## Abstract

Although Shapley additive explanations (SHAP) can be computed in polynomial time for simple models like decision trees, they unfortunately become NP-hard to compute for more expressive black-box models like neural networks — where generating explanations is often most critical. In this work, we analyze the problem of computing SHAP explanations for *Tensor Networks (TNs)*, a broader and more expressive class of models than those for which current exact SHAP algorithms are known to hold, and which is widely used for neural network abstraction and compression. First, we introduce a general framework for computing provably exact SHAP explanations for general TNs with arbitrary structures. Interestingly, we show that, when TNs are restricted to a *Tensor Train (TT)* structure, SHAP computation can be performed in *poly-logarithmic* time using *parallel* computation. Thanks to the expressiveness power of TTs, this complexity result can be generalized to many other popular ML models such as decision trees, tree ensembles, linear models, and linear RNNs, therefore tightening previously reported complexity results for these families of models. Finally, by leveraging reductions of binarized neural networks to Tensor Network representations, we demonstrate that SHAP computation can become *efficiently tractable* when the network's *width* is fixed, while it remains computationally hard even with constant *depth*. This highlights an important insight: for this class of models, width — rather than depth — emerges as the primary computational bottleneck in SHAP computation.

## 1 Introduction

Shapley additive explanations (SHAP) [75] represent a widely adopted method for obtaining post-hoc explanations for decisions made by ML models. However, one of its main limitations lies in its *computational intractability* [22, 106, 82]. While some frameworks mitigate this intractability using sampling heuristics or approximations, such as KernelSHAP [75, 42], FastSHAP [63], DeepSHAP [37], Monte Carlo-based methods [31, 116, 85, 50, 51, 32], or leverage score sampling [88], others focus on *exact* SHAP computations, by leveraging structural properties of certain model families to derive polynomial-time algorithms, such as the popular TreeSHAP [76] algorithm, and its variants [114, 84, 112, 86].

However, exact SHAP frameworks are confined to relatively simple models, such as tree-based models. For more expressive models like neural networks, where explanations are often most critical,

---

[*]Equal Contribution

NP-Hardness results have unfortunately been shown to hold [106, 82]. In this work, we conduct a complexity-theoretic analysis of the problem of computing exact SHAP explanations for the class of *Tensor Networks (TNs)*, a broader and more expressive class of models compared to previously studied models such as tree-based models or linear models.

**Tensor Networks.** Originally introduced in the field of Quantum physics [24], TNs have gradually garnered attention of the ML community where they were employed to model various ML tasks, ranging from classification and regression tasks [89, 101, 98, 34, 36, 53] to probabilistic modeling [83, 93, 102, 96, 52, 90], dimensionality reduction [77] and model compression [54]. From the perspective of explainable AI, the study of TNs is interesting in two ways. First, TNs offer a powerful modeling framework with function approximation capabilities comparable to certain families of neural networks [4, 94]. Second, the structured physics-inspired architecture of TNs enables the derivation of efficient, theoretically grounded solutions of XAI-related problems [73, 95, 3]. These two properties make TNs an interesting family of models that enjoy two arguably sought-after desiderata of ML models in the field of explainability: (i) high expressiveness; and (ii) enhanced transparency [97, 69].

**Contributions.** In this work, we provide a computational complexity-theoretic analysis of the problem of SHAP computation for the family of TNs. Following the line of work in [76, 6, 115, 66, 81, 82, 58] on tractable SHAP computation for simpler models, our main technical contributions are as follows:

1. **SHAP for general TNs.** We introduce a framework for computing provably exact SHAP explanations for general TNs with arbitrary structure, offering the first exact algorithm for generating SHAP values in this model class.

2. **SHAP for Tensor Trains (TTs).** We provide a deeper computational study of the SHAP problem for the particular class of *Tensor Trains (TTs)*, a popular subfamily of TNs that exhibits better tractability properties than general TNs. Interestingly, we show that computing SHAP for the family of TTs is not only computable in polynomial-time, but also belongs to the complexity class NC, i.e. it can be computed in *poly-logarithmic* time using *parallel* computation. This complexity result bridges a significant expressivity gap by establishing the tractability of SHAP computation for a model family that is significantly more expressive than those previously known to admit exact SHAP algorithms, while also demonstrating its efficient parallelizability.

3. **From TTs to improved complexity bounds for SHAP across additional ML models.** Via reduction to TTs, we show that the NC complexity result can be extended to the problem of SHAP computation for a wide range of additional popular ML models, including tree ensembles, decision trees, linear RNNs, and linear models, across various distributions, thus tightening previously known complexity results. This advancement benefits SHAP computation for these models in two key ways: (i) it enables substantially more efficient computation through poly-logarithmic parallelism, and (ii) it broadens the class of distributions used to compute SHAP's expected value, which captures more complex feature dependencies than those employed in current implementations.

4. **From TTs to SHAP for Binarized Neural Networks (BNNs).** Finally, through reductions from TTs, we reveal new complexity results for computing SHAP for BNNs via *parameterized complexity* [48, 46], a framework for assessing how structural parameters impact computational hardness. We find that while SHAP remains hard when *depth* is fixed, it becomes polynomial-time computable when *width* is bounded — highlighting width as the main computational bottleneck. We then further strengthen this insight by proving that fixing both width and *sparsity* (via the reified cardinality) renders SHAP *efficiently tractable*, even for arbitrarily large networks. This opens the door to a new, relaxed class of neural networks that permit efficient SHAP computation.

Beyond these core complexity results, which form the central focus of our work, our contributions also shed light into two novel complexity-theoretic aspects of SHAP computation that, to the best of our knowledge, have not been explored in prior literature:

**When is computing SHAP efficiently parallelizable?** To the best of our knowledge, this is the first work to provide a complexity-theoretic analysis of the computational *parallelizability* of SHAP computation. Our work provides a tighter complexity bound for many previously known tractable SHAP configurations by investigating conditions under which SHAP computation can be parallelized to achieve polylogarithmic-time complexity. Notably, we show that this is achievable for several

classical ML models, including decision trees, tree ensembles, linear RNNs, and linear models —
paving the way for a line of future research in this direction.

**What is the computational bottleneck of SHAP computation for neural networks?** To the best of
our knowledge, while prior work has established that computing SHAP for general neural networks
is computationally hard [106, 82], our work presents the first *fine-grained analysis* of how different
structural parameters influence this complexity. Focusing on binarized neural networks, we show that
SHAP becomes efficiently computable when both the network's width and sparsity are fixed, whereas
it remains hard even with constant depth. We believe this insight paves the way for further exploration
of neural network relaxations that enable efficient SHAP computation and invites broader theoretical
investigation into other structural parameters and architectures where SHAP may be tractable.

Due to space constraints, we include only brief outlines of some proofs in the main text, with complete
proofs provided in the appendix.

## 2 Preliminaries

**Notation.** For integers $(i, n)$ with $i \le n$, let $e_i^n$ denote the one-hot vector of length $n$ with a 1 in the
$i$-th position and 0 elsewhere. The vector $1_n \in \mathbb{R}^n$ denotes the vector equal to 1 everywhere. For
integers $m$ and $n$, we use the notation $m^{\otimes n} \stackrel{\text{def}}{=} \underbrace{[m] \times \ldots \times [m]}_{n \text{ times}}$ where $[m] = \{1, 2, \ldots, m\}$.

**Complexity classes.** In this work, we will assume familiarity with standard complexity classes such as
polynomial time (P), and nondeterministic polynomial time (NP and coNP). We further analyze the
complexity class #P, which captures the number of accepting paths of a nondeterministic polynomial-
time Turing machine, and is generally regarded as significantly "harder" than NP [7]. Moreover,
we analyze the complexity class NC, which includes problems solvable in *poly-logarithmic* time
using a polynomial number of *parallel* processors, typically on a Parallel Random Access Machine
(PRAM) [39] (see the appendix for a full formalization). Intuitively, a problem is in NC if it can
be *efficiently solved in parallel*. While NC $\subseteq$ P is known, it is widely believed that NC $\subsetneq$ P [7].
The class NC is further divided into subclasses NC$^k$ for some integer $k$. The parameter $k$ designates
the logarithmic order of circuits that can solve computational problems in NC$^k$. For example, NC$^1$
contains problems solvable with circuits of logarithmic depth, while NC$^2$ allows circuits of quadratic
logarithmic depth, capturing slightly more complex parallel computations.

Finally, our work also draws on concepts from *parameterized complexity* theory [46, 48]. In this
framework, problems are evaluated based on two inputs: the main input size $n$ and an additional
measure known as the *parameter* $k$, with the aim of confining the combinatorial explosion to $k$
rather than $n$. We focus on the three most commonly studied parameterized complexity classes:
(i) FPT (Fixed-Parameter Tractable), comprising problems solvable in time $g(k) \cdot n^{O(1)}$ for some
computable function $g$, implying tractability when $k$ is small; (ii) XP (slice-wise Polynomial), where
problems can be solved in time $n^{g(k)}$, with a polynomial degree that may grow with $k$, thus offering
weaker tractability guarantees than FPT; and (iii) para-NP, which captures problems with the highest
sensitivity to $k$, where a problem is para-NP-hard if it remains NP-hard even when $k$ is fixed to a
*constant*. It is widely believed that FPT $\subsetneq$ XP $\subsetneq$ para-NP [46].

**Shapley values.** Let $n_{in}, n_{out} \ge 1$ be two integers. Fix a discrete input space $\mathbb{D} = [N_1] \times [N_2] \times$
$\cdots \times [N_{n_{in}}]$, a model $M : \mathbb{D} \to \mathbb{R}^{n_{out}}$, and a probability distribution $P$ over $\mathbb{D}$. For an input instance
$x = (x_1, x_2, \ldots, x_{n_{in}}) \in \mathbb{D}$, the SHAP attribution vector assigned to the feature $i \in [n_{in}]$ is defined
as:

$$\phi_i(M, x, P) = \sum_{S \subseteq [n_{in}] \setminus \{i\}} W(S) \cdot \Big( V_M(x, S \cup \{i\}; P) - V_M(x, S; P) \Big) \in \mathbb{R}^{n_{out}} \qquad (1)$$

where $W(S) \stackrel{\text{def}}{=} \frac{|S|!(n_{in} - |S| - 1)!}{n_{in}!}$. We assume the common *marginal* (or *interventional*) value function:
$V_M(x, S; P) := \mathbb{E}_{x' \sim P}\Big[ M\big(x_S, x'_{\bar{S}}\big)\Big]$ [62, 103]. For $j \in [n_{out}]$, the $j$-th component of $\phi_i(M, x, P)$
represents the attribution of feature $i$ to the $j$-th output of the model on input $x$.

# 3 Tensors, Tensor Networks and Binarized Neural Networks

## 3.1 Tensors

A tensor is a multi-dimensional array that generalizes vectors and matrices to higher dimensions, referred to as *indices*. The dimensionality of a tensor, i.e. the number of its indices, defines its *order*. Elements of a tensor $\mathcal{T} \in \mathbb{R}^{d_1 \times \dots \times d_n}$ are denoted $\mathcal{T}_{i_1,\dots,i_n}$. For $j \in [n]$, the slice $\mathcal{T}_{i_1,\dots,i_{j-1},:,i_{j+1},\dots,i_n}$ is a vector in $\mathbb{R}^{d_j}$ whose $k$-th entry is equal to $\mathcal{T}_{i_1,\dots,i_{j-1},k,i_{j+1},\dots,i_n}$ for $k \in [d_j]$. A key operation over tensors is *the contraction operation*, which generalizes matrix multiplication to high-order tensors. Formally, given two tensors $\mathcal{T}^{(1)} \in \mathbb{R}^{d_1 \times \dots \times d_n}$ and $\mathcal{T}^{(2)} \in \mathbb{R}^{d'_1 \times \dots \times d'_m}$, and two indices $(i,j) \in [n] \times [m]$, such that $d_i = d'_j$ The contraction operation between $\mathcal{T}^{(1)}$ and $\mathcal{T}^{(2)}$ over their respective indices $i$ and $j$ produces another tensor, denoted $\mathcal{T}^{(1)} \times_{(i,j)} \mathcal{T}^{(2)}$, over $\mathbb{R}^{d_1 \times \dots d_{i-1} \times d_{i+1} \times \dots \times d_n \times d'_1 \times \dots \times d'_{j-1} \times d'_{j+1} \times \dots \times d'_m}$ such that:

$$\left( \mathcal{T}^{(1)} \times_{(i,j)} \mathcal{T}^{(2)} \right)_{a_1,\dots,a_{i-1},a_{i+1},\dots,a_n,b_1,\dots,b_{j-1},b_{j+1},\dots,b_m} \overset{\text{def}}{=\!=}$$
$$\sum_{k=1}^{d_i} \mathcal{T}^{(1)}_{a_1,\dots,a_{i-1},k,a_{i+1},\dots,a_m} \cdot \mathcal{T}^{(2)}_{b_1,\dots,b_{j-1},k,b_{j+1},\dots,b_n}$$

The contraction operation over a single pair of shared indices as defined above can be generalized to sets of shared indices in a natural fashion: For a set $S \subseteq [n] \times [m]$, the *multi-leg contraction operation* between $\mathcal{T}^{(1)}$ and $\mathcal{T}^{(2)}$ over $S$, denoted $\mathcal{T}^{(1)} \times_S \mathcal{T}^{(2)}$ applies leg contraction to all pairs of indices in $S$. Note that this operation is commutative — the result of the operation doesn't depend on the contraction order.

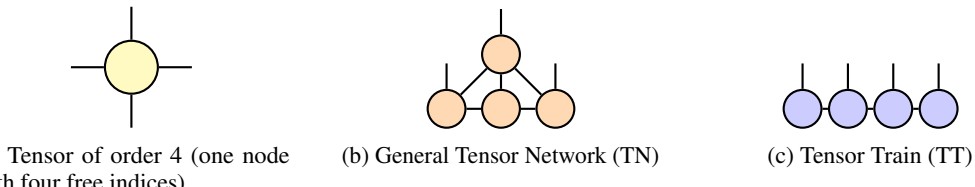

(a) Tensor of order 4 (one node with four free indices)    (b) General Tensor Network (TN)    (c) Tensor Train (TT)

Figure 1: Illustrations of (1) A tensor of order 4, (2) A general TN comprising 4 tensors of order 4 and 3 free indices, and (3) A tensor train (TT).

## 3.2 Tensor Networks (TNs)

TNs provide a structured representation of high-order tensors by decomposing them into interconnected collections of lower-order tensors bound by the contraction operation. TNs can be modeled as a graph in which each node corresponds to a tensor and each edge corresponds to an index. Edges that are incident to only one node represent free indices of the overall tensor, while edges shared between two nodes represent contracted indices (see Figure 5b). The tensor encoded by the network is obtained by carrying out all contractions prescribed by the graph structure. This representation of high-order tensors can be advantageous because the storage requirements and computational cost scale with the ranks of the intermediate tensors and the network topology, rather than with the full dimensionality of the original tensor.

**Tensor Trains (TTs).** General TNs with arbitrary topologies can be computationally challenging to handle [110, 100]. TTs form a subclass of TNs with a one-dimensional topology that exhibits better tractability properties [92]. A TT decomposes a high-order tensor into a linear chain of third-order "cores" (see Figure 5c). Formally, a TT $\mathcal{T}$ is a TN parametrized as $[\![\mathcal{T}^{(1)}, \dots, \mathcal{T}^{(n)}]\!]$, where each core $\mathcal{T}^{(i)} \in \mathbb{R}^{d_{i-1} \times N_i \times d_i}$. The TT $\mathcal{T}$ corresponds to a tensor in $\mathbb{R}^{d_0 \times N_1 \times \dots \times N_n \times d_n}$ obtained by contraction.

$$\mathcal{T} = \mathcal{T}^{(1)} \times_{(3,1)} \mathcal{T}^{(2)} \times_{(3,1)} \cdots \times_{(3,1)} \mathcal{T}^{(n)}$$

By convention, when $d_0$ (resp. $d_n$) is equal to 1, the tensor core $\mathcal{T}^{(1)}$ (resp. $\mathcal{T}^{(n)}$) is a matrix (i.e., a Tensor of order 2).

### 3.3 Binarized Neural Networks (BNNs)

A *Binarized Neural Network* (BNN) is a neural network in which both the weights and activations are constrained to binary values, typically $\{-1, +1\}$. In a BNN, each layer performs a sequence of operations: (i) a linear transformation, (ii) batch normalization, and (iii) binarization. Formally, for an input vector $\mathbf{x} \in \{-1, +1\}^n$, the output $\mathbf{y} \in \{-1, +1\}^m$ of a layer is computed as follows [41]:

$$\mathbf{z} = \mathbf{W} \cdot \mathbf{x} + \mathbf{b}, \quad \mathbf{z}' = \gamma \cdot \frac{\mathbf{z} - \mu}{\sigma} + \beta, \quad \mathbf{y} = \mathrm{sign}(\mathbf{z}'),$$

$\mathbf{W} \in \{-1, +1\}^{m \times n}$, $\mathbf{b} \in \mathbb{R}^m$, and $\gamma, \beta, \mu, \sigma \in \mathbb{R}^m$ are the batch normalization parameters. The sign function is applied element-wise, mapping positive inputs to $+1$ and non-positive inputs to $-1$.

**Reified Cardinality Representation of BNNs.** In Section 6, we analyze how network sparsity affects the complexity of computing SHAP for BNNs, using the *reified cardinality parameter* from [64]. A reified cardinality constraint is a binary condition of the form $(\sum_{i=1}^{n} x_i \geq R)$, where $x_i$ are Boolean variables and $R$ is the reified cardinality parameter. In BNNs, each neuron is encoded with such a constraint: for binary inputs $x_j$ and weights $w_{ij} \in \{-1, +1\}$, define $l_{ij} = x_j$ if $w_{ij} = +1$, and $l_{ij} = \neg x_j$ otherwise. The output $y_i$ is then: $y_i \leftarrow \left( \sum_{j=1}^{n} l_{ij} \geq R_i \right)$ where $R_i$ is derived from the neuron's bias and normalization. Jia and Rinard [64] showed that all BNNs can be expressed this way, and that constraining $R_i$ during training improves formal verification efficiency.

## 4 Provably Exact SHAP Explanations for TNs: A General Framework

In this section, we introduce a general framework for computing provably exact SHAP explanations for TNs with arbitrary structures, which provides the backbone of more tractable SHAP computational solutions. We begin by formalizing the problem and transitioning from the classic SHAP formulation to an equivalent *tensorized* representation that facilitates our proofs.

**A tensorized representation of Shapley values.** We adopt an equivalent *tensorized* form of the SHAP formula in Equation (1) by defining the Marginal SHAP Tensor $\mathcal{T}^{(M,P)} \in \mathbb{R}^{\mathbb{D} \times n_{in} \times n_{out}}$ as:

$$\forall (x, i) \in \mathbb{D} \times [n_{in}] : \quad \mathcal{T}^{(M,P)}_{x,i,:} \overset{\text{def}}{=} \phi_i(\mathcal{T}^M, x, \mathcal{T}^P) \tag{2}$$

The Marginal SHAP Tensor summarizes the full SHAP information of the model $M$, and will play a crucial role in the technical development of this paper. Given $\mathcal{V}^{(M,P)}$, The SHAP Matrix of the input $x = (x_1, \dots, x_{n_{in}}) \in \mathbb{D}$ is obtained using the straightforward contraction operation:

$$\Phi(\mathcal{T}^M, x, \mathcal{T}^P) = \mathcal{T}^{(M,P)} \times_S \left[ e_{x_1}^{N_1} \circ \dots \circ e_{x_{n_{in}}}^{N_{n_{in}}} \right] \tag{3}$$

where $S \overset{\text{def}}{=} \left\{ (k, k) : k \in [n_{in} + 1] \right\}$.

---

**SHAP Computation (Problem Statement).**
Given two integers $n_{in}$, $n_{out} \geq 1$, a finite domain $\mathbb{D}$, a TN model $\mathcal{T}^M \in \mathbb{R}^{N_1 \times \dots \times N_{n_{in}} \times n_{out}}$ mapping $\mathbb{D}$ to $\mathbb{R}^{n_{out}}$, a TN $\mathcal{T}^P \in \mathbb{R}^{N_1 \times \dots \times N_{n_{in}}}$ implementing a probability distribution over $\mathbb{D}$, and an instance $x \in \mathbb{D}$. The objective is to construct the SHAP Matrix $\Phi(\mathcal{T}^M, x, \mathcal{T}^P) \in \mathbb{R}^{n_{in} \times n_{out}}$, where: $\Phi(\mathcal{T}^M, x, \mathcal{T}^P)_{i,:} \overset{\text{def}}{=} \phi_i(\mathcal{T}^M, x, \mathcal{T}^P)$ (Equation (1)).

---

The complexity is measured in terms of the input and output dimensions $n_{in}$, $n_{out}$, the volume of the input space $\max_{i \in [n_{in}]} N_i$, and the total number of parameters of $\mathcal{T}^M$ and $\mathcal{T}^P$.

The first step toward the derivation of exact SHAP explanations for general TNs builds on a reformulation of the Marginal SHAP Tensor (Equation (2)) highlighted in the following proposition:

**Proposition 1.** *Define the* modified Weighted Coalitional tensor $\tilde{\mathcal{W}} \in \mathbb{R}^{n_{in} \times 2^{\otimes n_{in}}}$ *such that* $\forall (i, s_1, \dots, s_{n_{in}}) \in [n_{in}] \times [2]^{\otimes n_{in}}$ *it holds that* $\tilde{\mathcal{W}}_{i, s_1, \dots, s_{n_{in}}} \overset{\text{def}}{=} -W(s - 1_{n_{in}})$ *if* $s_i = 1$, *and* $\tilde{\mathcal{W}}_{i, s_1, \dots, s_{n_{in}}} \overset{\text{def}}{=} W(s - 1_{n_{in}})$ *otherwise. Moreover, define the* marginal value tensor $\mathcal{V}^{(M,P)} \in$

---

**Algorithm 1** The construction of the Marginal SHAP Tensor

---

**Input:** Two TNs $\mathcal{T}^M$ and $\mathcal{T}^P$
**Output:** The Marginal SHAP Tensor $\mathcal{T}^{(M,P)}$
  1: Construct the modified Weighted Coalitional Tensor $\tilde{\mathcal{W}}$ (Lemma 1)
  2: Construct the Marginal Value Tensor $\mathcal{V}^{(M,P)}$ (Lemma 2)
  3: **return** $\tilde{\mathcal{W}} \times_S \mathcal{V}^{(M,P)}$ (Equation (12))

---

$\mathbb{R}^{\mathbb{D} \times 2^{\otimes n_{in}} \times n_{out}}$, *such that* $\forall (x,s) \in \mathbb{D} \times [2]^{\otimes n}$ *it holds that* $\mathcal{V}^{(M,P)}_{x,s,:} \stackrel{\text{def}}{=} V_M(x,s;P)$. *Then we have that:*

$$\mathcal{T}^{(M,P)} = \tilde{\mathcal{W}} \times_S \mathcal{V}^{(M,P)} \tag{4}$$

*where* $S \stackrel{\text{def}}{=} \left\{ (k+1, k+n_{in}+1) : \ k \in [n_{in}] \right\}$.

Proposition 1 expresses the Marginal SHAP Tensor $\mathcal{T}^{(M,P)}$ as a contraction of two tensors: the *modified Weighted Coalitional Tensor* and the *Marginal Value Tensor*. This leads to a natural algorithm to construct $\mathcal{V}^{(M,P)}$: First construct $\tilde{\mathcal{W}}$ and $\mathcal{V}^{(M,P)}$, then contract them as in Equation (12) (Algorithm 1). Figure 2a illustrates the resulting TN of this process.

To complete the picture, we need to show how to construct both tensors $\tilde{\mathcal{W}}$ (Step 1) and $\mathcal{V}^{(M,P)}$ (Step 2). We split the remainder of this section into two segments, each of which is dedicated to outlining their structure and the running time of their construction.

**Step 1: Constructing the modified Weighted Coalitional Tensor.** The Tensor $\tilde{\mathcal{W}}$ simulates the computation of weights associated with each subset of the input features in the SHAP formulation (Equation (1)). A key observation consists of noting that this tensor admits an efficient representation as a (sparse) TT constructible in $O(1)$ time using parallel processors:

**Lemma 1.** *The modified Weighted Coalitional Tensor $\tilde{\mathcal{W}}$ admits a TT representation:* $[\![\mathcal{G}^{(1)}, \ldots, \mathcal{G}^{(n_{in})}]\!]$, *where* $\mathcal{G}^{(1)} \in \mathbb{R}^{n_{in} \times 2 \times n_{in}^2}$, $\mathcal{G}^{(i)} \in \mathbb{R}^{n_i^2 \times 2 \times n_i^2}$ *for any* $i \in [2, n_{in}-1]$, *and* $\mathcal{G}^{(n_{in})} \in \mathbb{R}^{n_{in}^2 \times 2}$. *Moreover, this TT representation is constructible in* $O(\log(n_{in}))$ *time using* $O\left(n_{in}^3\right)$ *parallel processors.*

**Step 2: Constructing the Marginal Value Tensor.** The goal of the second step is to construct the Marginal Value Tensor $\mathcal{V}^{(M,P)}$ from the TNs $\mathcal{T}^M$ and $\mathcal{T}^P$. The following lemma shows how this Tensor can be constructed by means of suitable TN contractions:

**Lemma 2.** *Let $\mathcal{T}^M$ and $\mathcal{T}^P$ be TNs implementing the model $M$ and the probability distribution $P$, respectively. Then, the marginal value tensor $\mathcal{V}^{(M,P)}$ can be computed as:*

$$\left[ \mathcal{M}^{(1)} \circ \ldots \circ \mathcal{M}^{(n_{in})} \right] \times_{S_1} \mathcal{T}^M \times_{S_2} \mathcal{T}^P$$

*where $S_1$ and $S_2$ are instantiated such that for all $k \in \{1,2\}$ it holds that:* $S_k \stackrel{\text{def}}{=} \left\{ (5-k) \cdot i, i \right) : i \in [n_{in}]\}$, *and for any $i \in [n_{in}]$, the (sparse) tensor $\mathcal{M}^{(i)} \in \mathbb{R}^{N_i \times 2 \times N_i^{\otimes 2}}$ is constructible in $O(1)$ time using $O(N_i^2)$ parallel processors.*

The collection of tensors $\{\mathcal{M}^{(i)}\}_{i \in [n_{in}]}$ can be interpreted as *routers* simulating the interventional mechanism in the Marginal SHAP formulation: Depending on the value assigned to its third index (which is binary), it routes the feature value of either its value in the input instance to explain $x$ or from one drawn from the data generating distribution $\mathcal{T}^P$ to feed the model's input.

## 5  Provably Exact and *Tractable* SHAP for TTs, and Other ML Models

In this section, we show that SHAP values can be computed in polynomial time for the specific case of Tensor Train (TT) models. Interestingly, we prove that the problem also lies in NC, meaning it can also be solved in *polylogarithmic* using parallel computation. This result is established in Subsection 5.1. Then, in Subsection 5.2, we demonstrate how this finding can be leveraged — via reduction — to tighten complexity bounds for several other popular ML models.

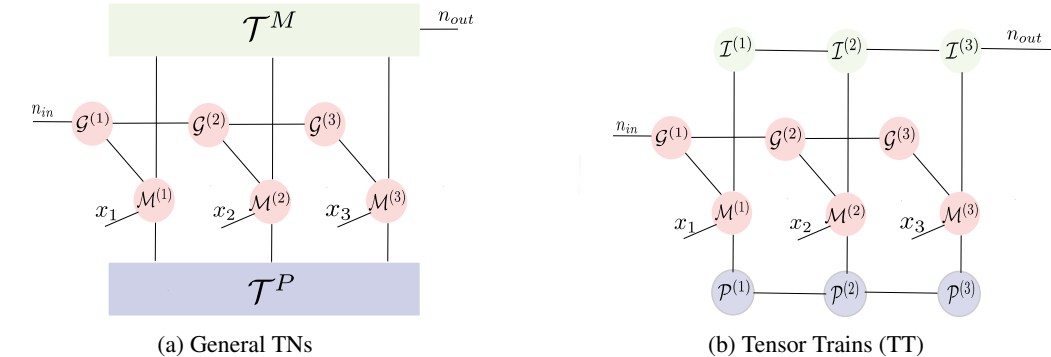

(a) General TNs

(b) Tensor Trains (TT)

Figure 2: The construction of the $\mathcal{T}^{(M,P)}$ TN for a model of 3 features. *The general case:* Both $\mathcal{T}^M$ and $\mathcal{T}^P$ are general TNs with arbitrary structures; *The TT case:* Both $\mathcal{T}^M$ and $\mathcal{T}^P$ are TTs.

### 5.1 Provably exact and *tractable* SHAP explanations for TTs

The computational complexity of computing SHAP for general TNs, as discussed in the previous section, naturally depends on the structural properties of the TNs $\mathcal{T}^M$ and $\mathcal{T}^P$ implementing the model to explain and the data generating distribution, respectively. We begin by presenting a general worst-case negative result, showing that the problem is #P-Hard when *no structural constraints* are imposed on the tensor networks.

**Proposition 2.** *Computing Marginal SHAP values for general TNs is #P-Hard.*

*Proof Sketch.* The result is obtained by reduction from the #CNF-SAT problem (The problem of counting the number of satisfying assignments of a CNF boolean formula ). Essentially, it leverages two facts: *(i)* the model counting problem is polynomially reducible to the SHAP problem [66]. *(ii)* A polynomial-time algorithm to construct an equivalent TN to a given CNF boolean formula. The full proof can be found in Appendix D.

Interestingly, the problem however, becomes tractable when we restrict both $\mathcal{T}^M$ and $\mathcal{T}^P$ to lie within the class of *Tensor Trains* (TTs). To establish this result, we begin with a key observation: the marginal SHAP tensor $\mathcal{V}^{(M,P)}$ itself admits a representation as a TT. Formally:

**Theorem 1.** *Let $\mathcal{T}^M = [\![ \mathcal{I}^{(1)}, \ldots, \mathcal{I}^{(n_{in})} ]\!]$ and $\mathcal{T}^P = [\![ \mathcal{P}^{(1)}, \ldots, \mathcal{P}^{(n_{in})} ]\!]$ be two TTs corresponding to the model to interpret and the data-generating distribution, respectively. Then, the Marginal SHAP Tensor $\mathcal{T}^{(M,P)}$ can be represented by a TT parametrized as:*

$$[\![ \mathcal{M}^{(1)} \times_{(4,2)} \mathcal{I}^{(1)} \times_{(3,2)} \mathcal{P}^{(1)} \times_{(2,2)} \mathcal{G}^{(1)}, \ldots \ldots, \mathcal{M}^{(n_{in})} \times_{(4,2)} \mathcal{I}^{(n_{in})} \times_{(3,2)} \mathcal{P}^{(n_{in})} \times_{(2,2)} \mathcal{G}^{(n_{in})} ]\!] \quad (5)$$

*where the collection of Tensors $\{ \mathcal{G}^{(i)} \}_{i \in [n_{in}]}$ and $\{ \mathcal{M}^{(i)} \}_{i \in [n_{in}]}$ are implicitly defined in Lemma 1 and Lemma 2, respectively.*

*Proof Sketch.* The result is obtained by plugging the TT corresponding to $\tilde{\mathcal{W}}$ (Lemma 1) and $\mathcal{V}^{(M,P)}$ (Lemma 2) into Equation 12, and performing a suitable arrangement of contraction ordering of the resulting TN. Figure 2b provides a visual description of how the TT structure of the SHAP Value Tensor $\mathcal{V}^{(M,P)}$ emerges when both $\mathcal{T}^M$ and $\mathcal{T}^P$ are TTs.

**Efficient parallel computation of SHAP for TTs.** Theorem 1 shows that computing exact SHAP values for TTs reduces to contracting the Marginal SHAP Tensor — a TT — with a rank-1 tensor representing the input (Equation 3). TT contraction is a well-studied problem with efficient parallel algorithms that run in poly-logarithmic time [83, 80], typically using a parallel scan over adjacent tensors. Leveraging this and the fact that matrix multiplication is in NC [28], we can prove the following:

**Proposition 3.** *Computing Marginal SHAP for the family of TTs lies in $\mathrm{NC}^2$.*

*Proof Sketch.* A parallel procedure to solve this problem runs as follows: First, compute in parallel tensor cores in Equation (20). Given that Matrix Multiplication is in $\mathrm{NC}^1$, this operation is also in $\mathrm{NC}^1$. Second, following [80], a parallel scan strategy will be applied to contract the resulting TT

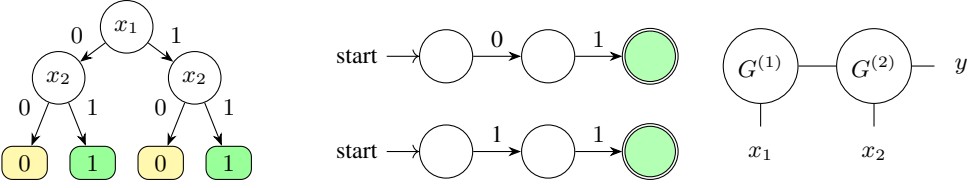

(a) Decision tree        (b) Lattice of finite-state machines        (c) Tensor Train representation

Figure 3: Conversion of a decision tree into an equivalent Tensor Train (TT). (a) A simple decision tree with two binary input features, (b) its equivalent lattice of finite-state automata, where each automaton encodes a distinct path leading to a leaf labeled $1$, and (c) the corresponding TT representation with three free legs: $x_1$, $x_2$,for inputs and $y$ for the output. The tensor cores $G^{(1)} \in \mathbb{R}^{2 \times 2}$ and $G^{(2)} \in \mathbb{R}^{2 \times 2 \times 2}$ have ranks equal to the number of automata in the lattice, i.e $\texttt{rank}(G^{(i)}) = 2$ for $i \in \{1, 2\}$

using a logarithmic depth of matrix multiplication operations. This second operation is performed by a circuit whose depth scales as $\mathcal{O}(\log^2(n_{in}))$ yielding the result (see Appendix D).

## 5.2 Tightening the complexity results of SHAP computations in many other ML models

Thanks to the expressive power of Tensor Trains (TTs), many widely used ML models — such as tree ensembles, decision trees, linear models, and linear RNNs — can be reduced to TTs. This reduction allows us to significantly tighten the known complexity bounds for these models. This improvement is crucial for two main reasons:

(i) It shows that computing SHAP values for all these models is not only polynomial-time solvable but also in the complexity class NC, enabling *efficient parallel computation*.

(ii) It demonstrates that Marginal SHAP can be computed under TT-based distributions — a class of distributions *more expressive* than those previously considered, englobing independent [6], empirical [106], Markovian [81], Hidden Markov distributions [82] and Born Machines [52].

We formalize this in the following theorem:

**Theorem 2.** *Computing Marginal SHAP values for decision trees, tree ensembles, linear models, and linear RNNs under the distribution class of TTs lies in* NC$^2$.

The proof of Theorem 2 can be found in Appendix E. It proceeds by constructing NC-reduction procedures that transform each of these ML models (i.e. Decision Trees, Tree Ensembles, Linear Models and Linear RNNs) into equivalent TTs.

For illustrative purposes, we show in figure 3 an example of such construction for a simple binary decision tree converted into an equivalent TT representation. The construction proceeds through an intermediate automata-based representation, where the DT is transformed into a lattice of finite state automata which admits a natural and compact parametrization in TT format. This construction can be implemented by means of a uniform family of boolean circuits with polynomial size and poly-logarithmic depth (see Appendix E for the formal details of this construction).

We believe that this result could be of interest to practitioners willing to scale the computation of SHAP explanations to large dimensions by leveraging parallelization under data generating distributions that capture sophisticated dependencies between input features.

## 6 A Fine-Grained Analysis of SHAP Computation for BNNs

In this section, we reveal an additional intriguing connection between our complexity results for TNs and Binarized Neural Networks (BNNs). This connection enables what is, to the best of our knowledge, the first *fine-grained analysis* of how particular structural parameters of a neural network affect the complexity of computing SHAP values.

We begin by noting that for a *non-quantized* neural network, even the seemingly simple case of a single sigmoid neuron with binary inputs has been shown to be intractable for SHAP computation [106, 82].

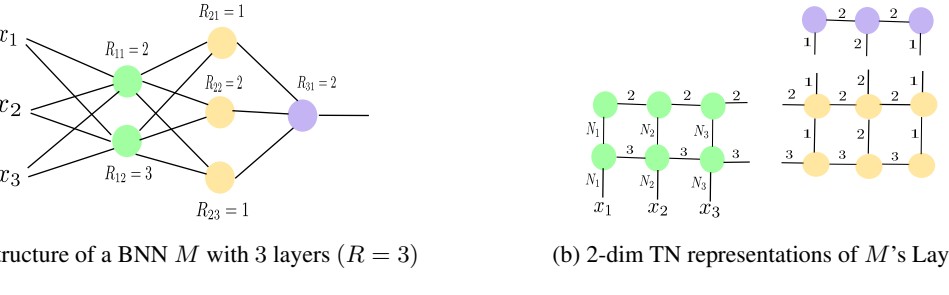

(a) Structure of a BNN $M$ with 3 layers ($R = 3$)   (b) 2-dim TN representations of $M$'s Layers

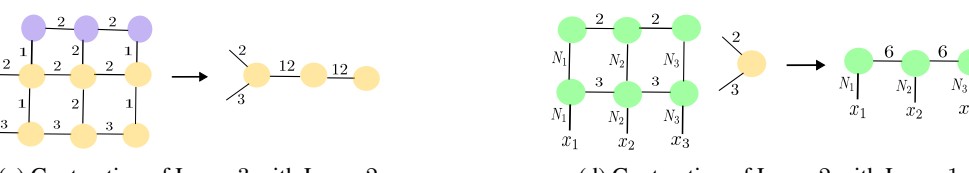

(c) Contraction of Layer 3 with Layer 2     (d) Contraction of Layer 2 with Layer 1

Figure 4: In Figure 4a, $R_{ij}$ denotes the reified cardinality parameters of the neurons. In Figures 4b, 4c, and 4d, the numbers above the edges indicate tensor index dimensionality.

In contrast, we show that by quantizing the weights and transitioning to a BNN, tractability *can* be achieved — though this tractability critically depends on the network's different structural parameters.

We carry out this analysis using the framework of *parameterized complexity* [48, 46], a standard approach in complexity theory for understanding how various structural parameters influence computational hardness. Specifically, we focus on three key parameters: (i) the network's *width*, (ii) its *depth*, and (iii) its *sparsity* . The main result of this analysis is captured in the following theorem:

**Theorem 3.** *Let $P$ be either the class of empirical distributions, independent distributions, or the class of TTs. We have that:*

1. ***Bounded Depth:*** *The problem of computing SHAP for BNNs under any distribution class $P$ is* PARA-NP-HARD *with respect to the network's* depth *parameter.*

2. ***Bounded Width:*** *The problem of computing SHAP for BNNs under any distribution class $P$ is in* XP *with respect to the* width *parameter.*

3. ***Bounded Width and Sparsity.*** *The problem of computing SHAP for BNNs under any distribution class $P$ is in* FPT *with respect to the* width *and* reified cardinality *parameters.*

**Key takeaways from these fine-grained results.** Theorem 3 captures several notable insights into the complexity of computing exact SHAP values for BNNs:

1. **Even shallow networks are hard.** Computing SHAP values for BNNs remains NP-Hard even when the network depth is fixed to a *constant*. In fact, intractability arises already with a *single* hidden layer, underscoring that reducing the network's depth does not alleviate the computational hardness of SHAP.

2. **Narrow networks make it easier.** However, when the *width* of the neural network is fixed to a constant, computing SHAP for a BNN becomes *polynomial* (as it falls within the class XP), highlighting *width* as a potential relaxation point for improving tractability.

3. **Narrow and sparse networks are efficient.** Finally, by fixing both the network's *width* and *sparsity*, we obtain an even stronger result: *fixed-parameter tractability (FPT)*. This implies that computing SHAP is *efficiently tractable* — even for arbitrarily large networks — so long as these parameters remain small, regardless of the network's depth, input dimensionality, or number of non-linear activations.

This shows that *width* drives the shift from intractability to tractability in SHAP for BNNs, and that adding sparsity bounds can fully tame complexity — even in high-dimensional settings.

*Proof Sketch (Theorem 3).* The first part follows from a direct reduction from 3SAT: any CNF formula can be encoded by a depth-2 BNN, and computing SHAP for CNF formulas is #P-Hard [106].

The second and third parts rely on a more involved construction: converting a BNN into an equivalent Tensor Train (TT), inspired by [72]. Each BNN layer is represented as a 2D tensor network, and these are contracted backwards into a TT. Figure 4 illustrates this process. The compilation of a BNN into an equivalent TT runs in $\mathcal{O}(R^W \cdot \texttt{poly}(D, n_{in}, \max_i N_i))$ time, where $W$, $D$, and $R$ are the width, depth, and reified cardinality of the BNN. The last two items of Theorem 3 follows immediately from this runtime complexity by definition of XP and FPT (see section 2), and the fact that SHAP for TTs is in NC (Proposition 3).

# 7   Limitations and Future Work

While our work represents a notable step toward understanding the computational landscape of SHAP, it remains focused on specific settings, and many other settings could naturally be explored. Future research could extend our analysis to additional model classes, such as Tree Tensor Networks [35], SHAP variants [103], and relaxations or approximations that enhance tractability for even more expressive models than those studied here. We also acknowledge existing critiques of SHAP [70, 57, 99, 49, 25]; our goal is not to defend its axiomatic foundations but to analyze the complexity of a widely adopted explanation method. Exploring the complexity of obtaining alternative value function definitions [71], SHAP variants [82], or other attribution indices [10, 113] presents exciting directions for future work.

# 8   Conclusion

In this work, we present the first provably exact algorithm for computing SHAP explanations for the class of *Tensor Networks*. Moreover, we prove that for the particular subclass of *Tensor Trains*, this computation can be carried out not only in polynomial time but also in polylogarithmic time using parallel processors. This result closes a *significant expressivity gap* in tractable SHAP computation by extending tractability to a significantly more expressive model family than previously known. Building on this expressivity, our approach also yields new insights into the complexity of computing SHAP for other popular ML models — including decision trees, linear models, linear RNNs, and tree ensembles — by enabling improved parallelizability-related complexity bounds and more expressive distribution modeling. Furthermore, these results offer, by reduction, a novel *fine-grained* analysis of the tractability barriers in computing SHAP for binarized neural networks, identifying *width* as a central computational bottleneck. Together, we believe that these findings significantly advance our understanding of the computational landscape of SHAP, highlighting both inherent limitations and new opportunities, and hence paving the way for future research.

## Acknowledgments

This work was partially funded by the European Union (ERC, VeriDeL, 101112713). Views and opinions expressed are however those of the author(s) only and do not necessarily reflect those of the European Union or the European Research Council Executive Agency. Neither the European Union nor the granting authority can be held responsible for them. This research was additionally supported by a grant from the Israeli Science Foundation (grant number 558/24).

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

# Appendix

The appendix is organized as follows:

**Appendix A** provides extended related work that is relevant to this work.

**Appendix B** provides the technical background, including descriptions of the model families studied in this work, as well as other technical tools — such as Copy Tensors and Layered Deterministic Finite Automata (LDFAs) — used in the proofs of various results presented in the paper.

**Appendix C** provides proofs related to the exact SHAP computation for Tensor Networks with arbitrary structures, namely Proposition 1, Lemma 1, and Lemma 2.

**Appendix D** provides the proof of the #P-Hardness of computing SHAP for general TNs (Proposition 2), along with a proof of the NC complexity result for the case of TTs (Proposition 3).

**Appendix E** presents the proof of Theorem 2, detailing the NC reductions from the SHAP computation problem for decision trees, tree ensembles, linear models, and linear RNNs to the corresponding SHAP problem for Tensor Trains (TTs).

**Appendix F** provides proofs on the fine-grained analysis of BNNs and their connection to TNs (Theorem 3).

## A    Extended Related Work

This section offers an expanded discussion of related work, with particular emphasis on key complexity results established in prior studies.

**SHAP values.** Building on the original SHAP framework introduced by [75] for explaining machine learning predictions, a substantial body of subsequent work has explored its application across a wide range of XAI settings. This includes the development of numerous SHAP variants [103, 62, 55], adaptations that align SHAP with underlying data manifolds [49, 105], and extensions that attribute higher-order feature interactions [50, 51, 104, 86]. From the computational perspective, a variety of statistical approximation techniques have been proposed to improve scalability [50, 104, 86, 87], alongside model-specific implementations for classes such as linear models [75], tree-based models [114, 84, 112, 86, 76], additive models [47, 21, 27], and kernel methods [32, 33]. Closer to our line of work, recent efforts aim to construct neural network architectures that support efficient computation of SHAP values [87, 38]. Finally, several parallel research threads have highlighted important limitations and potential failure modes of SHAP across different settings [49, 57, 70, 79].

**The computational complexity of SHAP.** Notably, [106] study SHAP when the value function is defined via conditional expectations and establish both tractability and intractability results across various ML model classes. Building on this, [6] generalize these findings, showing that tractability for SHAP using conditional expectations coincides precisely with the class of Decomposable Deterministic Boolean Circuits, and further prove that both decomposability and determinism are necessary for tractable computation. More recently, [81] extend the analysis beyond independent feature distributions to Markovian distributions, which was later generalized to HMM-distributed features [82], alongside an exploration of additional SHAP variants — such as baseline and marginal SHAP — as well as both local and global computation settings. Furthermore, [58, 82] distinguish between the complexity of SHAP for regression versus classification tasks. Other, more tangential but relevant extensions of these complexity studies include analyzing SHAP computations in database settings [45, 74, 23, 66, 67], as well as investigating the complexity of alternative attribution schemes beyond Shapley values — such as Banzhaf values [1] and other cooperative game-theoretic values [10]. In contrast, our work significantly broadens existing complexity results by considering much more expressive classes for both the underlying prediction models and the data distributions. Moreover, we are the first to provide a complexity-theoretic analysis of the *parallelizability* of SHAP, as well as fine-grained complexity bounds for computing SHAP values in families of neural networks.

**Formal XAI.** More broadly, this work lies within the emerging area of *formal explainable AI* (formal XAI) [78, 14], which seeks to derive explanations for machine learning models that come with mathematical guarantees [59, 14, 43, 44, 18, 60, 20, 8, 111, 65, 29]. Explanations in this setting are typically produced through automated reasoning techniques, such as SMT solvers [13] (e.g., for explaining tree ensembles [9]) and neural network verifiers [68, 109, 108] (e.g., for explaining neural networks [61, 15]). Since computing explanations with provable guarantees is often computationally intractable, a central line of work in formal XAI focuses on analyzing the computational complexity of explanation problems [11, 107, 40, 16, 26, 5, 19, 17, 2, 12, 30, 91].

## B  Background

This section begins by formally introducing the ML models examined in this work. It then presents the concept of copy tensors, followed by an introduction to Layered Deterministic Finite Automata (LDFAs). Both copy tensors and LDFAs are structures used to support several proofs throughout the appendix.

### B.1  Model Formalization

#### B.1.1  Decision Trees

We define a decision tree (DT) as a directed acyclic graph that represents a function $f : \mathbb{D} \to [c]$, where $c \in \mathbb{N}$ is the number of classes. The tree structure encodes this function as follows: (i) Each internal node $v$ is associated with a unique binary input feature from $\{1, \dots, n\}$; (ii) Each internal node has at most $k$ outgoing edges, corresponding to values in $[k]$ assigned to the feature at $v$; (iii) Along any path $\alpha$, each feature appears at most once; (iv) Each leaf node is labeled with a class from $[c]$. Given an input $x \in \mathbb{D}$, the DT defines a unique path $\alpha$ from the root to a leaf, where $f(x)$ is the class label at the leaf. The size of the DT, denoted $|f|$, is the total number of edges in the graph. To allow modeling flexibility, the order of variables can differ across paths $\alpha$ and $\alpha'$, but no variable repeats within a single path.

#### B.1.2  Decision Tree Ensembles

There are several well-established architectures for tree ensembles. Although these models mainly differ in how they are trained, our focus is on post-hoc interpretation, so we concentrate on differences in the inference phase. In particular, we analyze ensemble families that apply weighted-voting schemes during inference — this includes boosted ensembles such as XGBoost. When all weights are equal, our framework also captures majority-voting ensembles, like those used in bagging methods such as Random Forests. Conceptually, an ensemble tree model $\mathcal{T}$ is defined as a weighted sum of individual decision trees. Formally, it is specified by the tuple $\langle \{T_i\}_{i \in [m]}, \{w_i\}_{i \in [m]} \rangle$, where $\{T_i\}_{i \in [m]}$ denotes a collection of decision trees (the ensemble), and $\{w_i\}_{i \in [m]}$ is a set of real-valued weights. The model $\mathcal{T}$ is applied to regression tasks and the function it computes is defined as follows:

$$f_\mathcal{T}(x_1, \dots, x_n) := \sum_{i=1}^{m} w_i \cdot f_{T_i}(x_1, \dots, x_n) \tag{6}$$

#### B.1.3  Linear RNNs and Linear Regression Models.

Recurrent Neural Networks (RNNs) are neural models specifically designed to handle sequential data, making them ideal for tasks involving time-dependent or ordered information. They are widely used in applications such as language modeling, speech recognition, and time series prediction. A (second-order) linear RNN is a sub-class of RNNs, that models non-linear multiplicative interactions between the input and the RNN's hidden state. Formally, a second-linear RNN $R$ is parametrized by the tuple $< h_0, \mathcal{T}, W, U, b, O >$ where :

- $h_0 \in \mathbb{R}^d$: is the initial state vector.
- $\mathcal{T} \in \mathbb{R}^{d \times d \times d}$: Second-order interaction tensor capturing the multiplicative interactions between $x_t^{(i)}$ and $h_{t-1}^{(j)}$.

- $W \in \mathbb{R}^{d \times N}$: The input matrix
- $U \in \mathbb{R}^{d \times d}$: The state transition matrix
- $b \in \mathbb{R}^d$: The bias vector
- $O \in \mathbb{R}^{N \times n_{out}}$: The observation matrix

The parameter $d$ is referred to as the size of the RNN, and $N$ is the vocabulary size.

The processing of a vector $x = (x_1, \ldots, x_{n_{in}}) \in [N]^{n_{in}}$ by an RNN is performed sequentially from left to right according to the equation:

$$\begin{cases} \mathbf{h}_t = \mathcal{T} \times_{(1,1)} \mathbf{h}_{t-1} \times_{(2,1)} e_{x_t}^N + W e_{x_t}^N + U \mathbf{h}_{t-1} + b \\ \mathbf{y} = O^T \cdot \mathbf{h}_{n_{in}} \end{cases}$$

Where $\mathbf{h}_{t-1}$ is the hidden state vector of the RNN after reading the prefix $x_{1:y}$ of the input sequence, and $\mathbf{y}$ is the model's output for the input instance $x$.

Second-order linear RNNs, as defined above, generalize two classical ML models, namely *first-order linear RNNs* and *Linear Regression Models*. When the second-order interaction tensor $\mathcal{T}$ is set to the zero tensor, then the family of second-order linear RNNs is reduced to the family of first-order linear RNNs. If, in addition, the matrix $U$ is equal to the identity matrix, then a second-order linear RNN is reduced to a linear regression model.

We use three main tensor operations in this work:

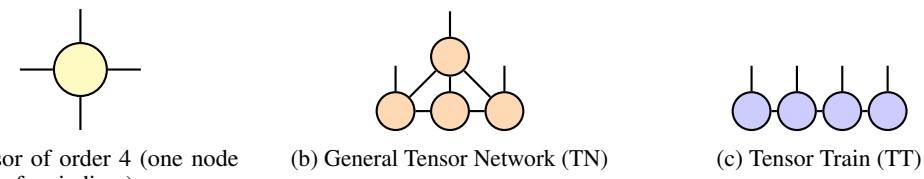

(a) Tensor of order 4 (one node with four free indices)  (b) General Tensor Network (TN)  (c) Tensor Train (TT)

Figure 5: Illustrations of (1) A tensor of order 4, (2) A general TN comprising 4 tensors of order 4 and 3 free indices, and (3) A tensor train (TT).

## B.2 Elements of Tensor Algebra: Operations over Tensors, Copy Tensors

### B.2.1 Operations over Tensors

In this article, we mainly use three operations over Tensors:

**Operation 1: Contraction.** Given two Tensors $\mathcal{T}^{(1)} \in \mathbb{R}^{d_1 \times \ldots \times d_n}$ and $\mathcal{T}^{(2)} \in \mathbb{R}^{d'_1 \times \ldots \times d'_m}$, and two indices $(i, j) \in [n] \times [m]$, such that $d_i = d'_j$ The contraction of Tensors $\mathcal{T}^{(1)}$ and $\mathcal{T}^{(2)}$ over indices $i$ and $j$ produces another tensor, denoted $\mathcal{T}^{(1)} \times_{(i,j)} \mathcal{T}^{(2)}$, over $\mathbb{R}^{d_1 \times \ldots d_{i-1} \times d_{i+1} \times \ldots \times d_n \times d'_1 \times \ldots \times d'_{j-1} \times d'_{j+1} \times \ldots \times d'_m}$ such that:

$$\left( \mathcal{T}^{(1)} \times_{(i,j)} \mathcal{T}^{(2)} \right)_{a_1, \ldots, a_{i-1}, a_{i+1}, \ldots, a_n, b_1, \ldots, b_{j-1}, b_{j+1}, \ldots, b_m} \stackrel{\text{def}}{=}$$

$$\sum_{k=1}^{d_i} \mathcal{T}^{(1)}_{a_1, \ldots, a_{i-1}, k, a_{i+1}, \ldots, a_m} \cdot \mathcal{T}^{(2)}_{b_1, \ldots, b_{j-1}, k, b_{j+1}, \ldots, b_n}$$

We slightly abuse notation and generalize single-leg contraction to *multi-leg contraction*. For a set $S \subseteq [n] \times [m]$, the multi-leg contraction $\mathcal{T}^{(1)} \times_S \mathcal{T}^{(2)}$ applies leg contraction to all index pairs in $S$. This operation is commutative — the result doesn't depend on contraction order.

**Operation 2: Outer Product.** Given two Tensors $\mathcal{T}^{(1)} \in \mathbb{R}^{d_1 \times \ldots \times d_n}$ and $\mathcal{T}^{(2)} \in \mathbb{R}^{d'_1 \times \ldots \times d'_m}$. The Tensor Outer product of $\mathcal{T}^{(1)}$ and $\mathcal{T}^{(2)}$ produces a Tensor, denoted $\mathcal{T}^{(1)} \circ \mathcal{T}^{(2)}$, in $\mathbb{R}^{d_1 \times \ldots \times d_n \times d'_1 \times \ldots \times d'_m}$ whose parameters are given as follows:

$$(\mathcal{T}^{(1)} \circ \mathcal{T}^{(2)})_{a_1, \ldots a_n, b_1, \ldots b_m} \stackrel{\text{def}}{=} \mathcal{T}^{(1)}_{a_1, \ldots, a_n} \cdot \mathcal{T}^{(2)}_{b_1, \ldots, b_m}$$

**Operation 3: Reshape.** Let $\mathcal{T} \in \mathbb{R}^{d_1 \times d_2 \times \cdots \times d_n}$ be a tensor of order $n$, and let $[m_1, m_2, \ldots, m_k]$ be a list of positive integers such that $m_1 + m_2 + \cdots + m_k = n$. Define: $p(i) \stackrel{\text{def}}{=} 1 + \sum_{l=1}^{i-1} m_l$, $q(i) \stackrel{\text{def}}{=} \sum_{l=1}^{i} m_l$ and $D_i \stackrel{\text{def}}{=} \prod_{j=p(i)}^{q(i)} d_j$. Then, the reshape operation, denoted $\texttt{reshape}(\mathcal{T}, [m_1, \ldots, m_k])$ produces a new tensor $\mathcal{T}' \in \mathbb{R}^{D_1 \times D_2 \times \cdots \times D_k}$ via a bijection between the multi-index $(i_1, \ldots, i_n)$ of $\mathcal{T}$ and $(J_1, \ldots, J_k)$ of $\mathcal{T}'$. For each group $i$, the combined index is: $J_i = 1 + \sum_{j=p(i)}^{q(i)} \left( (i_j - 1) \prod_{l=j+1}^{q(i)} d_l \right)$, and we set $\mathcal{T}'(J_1, \ldots, J_k) = \mathcal{T}(i_1, \ldots, i_n)$. This mapping is bijective and preserves the number of elements.

### B.2.2  Copy Tensors

In the field of TNs, a copy tensor, also known as a delta tensor or Kronecker tensor [24], is a special type of tensor that enforces equality among its indices. It ensures that all connected indices must take the same value, making it essential for modeling constraints where multiple indices must be identical. This tensor is widely used in operations requiring synchronized index values across different parts of a network. In our context, it will be employed to simulate BNNs using TNs (Section F), as well as in the proof of the #P-Hardness of computing Marginal SHAP for General TNs (subsection D.1). Formally, for $n \geq 1$, the elements of the order-$n$ copy tensor, denoted $\Delta^{(n)}$, defined over the index set $[d]$ are given by:

$$\Delta^{(n)}_{i_1, i_2, \ldots, i_n} := \begin{cases} 1 & \text{if } i_1 = i_2 = \cdots = i_n \\ 0 & \text{otherwise} \end{cases} \tag{7}$$

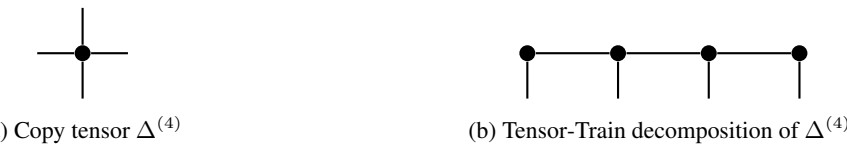

(a) Copy tensor $\Delta^{(4)}$        (b) Tensor-Train decomposition of $\Delta^{(4)}$.

Figure 6: (a) The copy tensor $\Delta^{(4)}$ enforces index-equality among its $n$ legs. (b) Its exact TT representation is a chain of 4 tensor cores, each a black node with one "physical" leg (down) and two "bond" legs (horizontal), enforcing $i_1 = \cdots = i_n$.

In TN diagrams, a copy tensor is traditionally depicted as a black dot with $n$ edges, where each edge represents one of its indices (Figure 6).

**Copy Tensors as Tensor Trains (TTs).** The explicit construction of the copy tensor $\Delta^{(n)}$ scales exponentially with $n$. Fortunately, as shown in [72], the copy tensor $\Delta^{(n)}$ admits an exact length TT decomposition $n$.

$$\mathcal{G}^{(k)} \in \mathbb{R}^{r_{k-1} \times d \times r_k}, \qquad r_0 = r_n = 1, \quad r_k = d \ (1 \leq k \leq n-1) \tag{8}$$

With the following elements:

$$\mathcal{G}^{(k)}_{\alpha_{k-1}, i_k, \alpha_k} := [\![\alpha_{k-1} = i_k]\!] \cdot [\![\alpha_k = i_k]\!] \tag{9}$$

Where $\alpha_0 = \alpha_n = 1$, and each $\alpha_k, i_k \in \{1, \ldots, d\}$. Hence, we have the following:

$$\Delta^{(n)}_{i_1, \ldots, i_n} = \sum_{\alpha_1, \ldots, \alpha_{n-1}=1}^{d} \mathcal{G}^{(1)}_{1, i_1, \alpha_1} \, \mathcal{G}^{(2)}_{\alpha_1, i_2, \alpha_2} \, \cdots \, \mathcal{G}^{(n)}_{\alpha_{n-1}, i_n, 1}. \tag{10}$$

Since each core enforces $\alpha_{k-1} = i_k = \alpha_k$, the only nonzero term in the sum occurs when $i_1 = \cdots = i_n$, thereby exactly reproducing the copy tensor.

## B.3 Layered Deterministic Finite Automata (LDFAs).

A *Layered Deterministic Finite Automaton* (LDFA) is a restricted form of a deterministic finite automaton [56] with a strict layered structure. Let $\Sigma = [N]$ be a finite input alphabet. An LDFA over $\Sigma$ is defined as a tuple,

$$\mathcal{A} = \left(L, \{S_l\}_{l=1}^L, \Sigma, \delta, s_1^0, F_L\right)$$

where:

- $L \geq 1$ is the number of layers;
- For each $l \in [L]$, $S_l$ is a finite set of states at layer $l$, and the total set of states is $Q = \biguplus_{l=1}^L S_l$;
- $\delta$ is a partial transition function:

$$\delta : \{(s, \sigma) \mid s \in S_l, \sigma \in \Sigma, 1 \leq l < L\} \rightarrow S_{l+1}$$

  such that for each $(s, \sigma)$, $\delta(s, \sigma)$ is uniquely defined if a transition exists, and otherwise undefined;
- $s_1^0 \in S_1$ is the unique initial state, located in the first layer;
- $F_L \subseteq S_L$ is the set of accepting states, all located in the final layer.

The LDFA processes sequences of length $L$. A sequence $w = \sigma_1 \sigma_2 \cdots \sigma_L$ is accepted if there exists a sequence of states $s_1^0, s_2^1, \ldots, s_{L+1}^L$ such that

$$s_{l+1}^l = \delta(s_l^{l-1}, \sigma_l), \quad \text{for all } l = 1, \ldots, N,$$

and $s_{L+1}^L \in F_L$. By construction, transitions only go forward one layer at a time, making the automaton acyclic and enforcing a strict pipeline topology.

**LDFAs as TTs.** Layered Deterministic Finite Automata (LDFAs) admit a natural and efficient representation in the Tensor Train (TT) format. Given an LDFA $\mathcal{A} = \left(L, \{S_l\}_{l=1}^L, \Sigma, \delta, s_1^0, F_L\right)$ over the alphabet $\Sigma = [N]$, we construct a sequence of tensor cores $\{\mathcal{G}^{(l)}\}_{l=1}^L$, where each core $\mathcal{G}^{(l)}$ is a sparse binary 3-dimensional tensor encoding the transition map $\delta$ from layer $l$ to $l+1$. Each core $\mathcal{G}^{(l)}$ has dimensions corresponding to $|S_l| \times [N] \times |S_{l+1}|$, where an entry $\mathcal{G}^{(l)}(s, \sigma, s') = 1$ if and only if $\delta(s, \sigma) = s'$ (i.e. the transition from the state $s$ to the state $s'$ is valid after reading the inpit $\sigma$.

Using the TT representation of LDFAs, the acceptance of the input $x = (x_1, \ldots, x_L)$ can be alternatively computed as:

$$f(x) = \left[e_{s_1^0}^N \times_{(1,1)} \mathcal{G}^{(1)} \times_{(3,1)} \cdot G^{(2)} \times_{(3,1)} \cdots \times_{(3,1)} \mathcal{G}^{(L)} \cdot e_{F_L}^N\right] \times_S \left[e_{x_1}^N \circ \cdots \circ e_{x_L}^N\right]$$

where $S \overset{\text{def}}{=} \{(i, i) : i \in [L]\}$.

It's worth noting that the conversion from an LDFA to its TT representation can be performed in polynomial time with respect to the number of layers $L$ and the total number of states $\sum_{l=1}^L |S_l|$, since each transition is represented by a single non-zero entry in a sparse tensor core. This TT encoding enables the efficient representation and manipulation of the automaton using TN operations.

**Product Operation over LDFAs.** Given two LDFAs $\mathcal{A}^{(1)} = (L, \{S_l^1\}_{l=1}^L, \Sigma, \delta^1, s_1^{0,1}, F_L^1)$ and $\mathcal{A}^{(2)} = (L, \{S_l^2\}_{l=1}^L, \Sigma, \delta^2, s_1^{0,2}, F_L^2)$ with identical input alphabet and number of input layers, the product of $\mathcal{A}^{(1)}$ and $\mathcal{A}^{(2)}$, denoted $\mathcal{A}^{(1)} \times \mathcal{A}^{(2)}$, is a LDFA with the following parameterization:

$$\mathcal{A}^{(1)} \times \mathcal{A}^{(2)} = \left(L, \{S_l^1 \times S_l^2\}_{l=1}^L, \Sigma, \delta^1 \times \delta^2, (s_1^{0,1}, s_1^{0,2}), F_L^1 \times F_L^2\right)$$

where for any $l \in [L]$ and any pair of states $(s_l^1, s_l^2) \in S_l^1 \times S_l^2$, and $\sigma \in \Sigma$, we have:

$$(\delta^1 \times \delta^2)\left((s_l^1 \times s_l^2), \sigma\right) = \left(\delta_1(s_l^1, \sigma), \delta^2(s_l^2, \sigma)\right)$$

The function computed by the LDFA $\mathcal{A}^{(1)} \times \mathcal{A}^{(2)}$ is equal to $f_{\mathcal{A}^{(1)} \times \mathcal{A}^{(2)}} = f_{\mathcal{A}^{(1)}} \cdot f_{\mathcal{A}^{(2)}}$ [81]. Concretely, the product operation on LDFAs computes the intersection of two LDFAs. Equivalently,

when LDFAs encode clauses, this operation corresponds to the conjunction of those clauses. Note that the product operation over LDFAs is commutative.

In the context of our work, we are interested in efficient encodings of the product of multiple LDFAs into a single TN. Fix an integer $k \geq 1$ and let $\{\mathcal{A}^{(k)}\}_{k \in [K]}$ be $K$ LDFAs sharing the same input alphabet and the same number of layers. We aim at constructing a TN that computes the function: $f_{\mathcal{A}^{(1)}} \times \ldots f_{\mathcal{A}^{(K)}}$. If we restrict our TN representation to be in TT format, a naive construction of an equivalent TT consists of applying the Kronecker product over transition maps of all these LDFAs [81]. The state space of the resulting LDFA from this operation scales exponentially with respect to $K$. However, for general TNs, one can prove that the product over LDFAs admits an efficient encoding in polynomial time:

**Lemma 1.** *There exists a polynomial algorithm that takes as input a set of $K$ LDFAs $\{A^{(k)}\}_{k \in [K]}$ sharing the same input alphabet $\Sigma$, and the same number of layers $L$ and outputs a TN equivalent to $\mathcal{A}^{(1)} \times \ldots \times A^{(K)}$. The complexity is measured in terms of $K$, the alphabet size, and the number of states of the input LDFAs.*

The key building block for proving Lemma 1 is the copy tensor introduced in the previous section. Specifically, for each layer $l \in [L]$, the tensor obtained by contracting the copy tensor $\Delta^{(K)}$ with the $l$-th tensor cores of the TT representations of the LDFAs matches the transition map of the resulting LDFA produced by the product operation. We show this fact in the following proposition. To ease exposition, we restrict our result for the case $K = 2$. Thanks to the commutative property of the product operation over LDFAs, the general case follows by induction.

**Proposition 1.** *Let $\mathcal{A}^{(1)}$, and $\mathcal{A}^{(2)}$ be two LDFAs (sharing the same input alphabet and the same number of layers) parametrized in TT format as $[\![\mathcal{G}^{(1,1)}, \ldots, \mathcal{G}^{(1,L)}]\!]$ and $[\![\mathcal{G}^{(2,1)}, \ldots, \mathcal{G}^{(2,L)}]\!]$, respectively. Then, the TT is parametrized as:*

$$[\![\Delta^{(3)} \times_{(2,1)} \mathcal{G}^{(1,1)} \times_{(3,1)} \mathcal{G}^{(2,1)}, \ldots, \Delta^{(3)} \times_{(2,1)} \mathcal{G}^{(1,L)} \times_{(3,1)} \mathcal{G}^{(2,L)}]\!] \tag{11}$$

*is equivalent to $\mathcal{A}^{(1)} \times \mathcal{A}^{(2)}$.*

*Proof.* Let $\mathcal{A}^{(1)} = (L, \{S_l\}_{l=1}^L, \Sigma, \delta^1, s_1^{0,1}, F_L^1)$ and $\mathcal{A}^{(2)} = (L, \{S_l'\}_{l=1}^L, \Sigma, \delta^2, s_1'^{0,1}, F_L'^1)$ be two LDFAs sharing the same input alphabet $\Sigma$ and the same number of layers $L$. We show that at each layer $l \in [L-1]$, the TT whose parameterization is given in equation (11) simulates exactly the dynamics governed by the transition map of the product LDFA $\mathcal{A}^{(1)} \times \mathcal{A}^{(2)}$ at layer $l$, namely $\delta_l^1 \times \delta_l^2$.

More formally, denote $[\![\mathcal{G}^{(1,1)}, \ldots, \mathcal{G}^{(1,L)}$ and $[\![\mathcal{G}^{(2,1)}, \ldots, \mathcal{G}^{(2,L)}]\!]$ the TT parametrizations of $\mathcal{A}^{(1)}$ and $\mathcal{A}^{(2)}$, respectively. We need to show that, for any $(i_l, i_{l+1}, \sigma, j_l, j_{l+1}) \in [S_l] \times [S_{l+1}] \times \Sigma \times [S_l'] \times [S_{l+1}'])$, we have:

$$\left(\Delta^{(3)} \times_{(2,1)} \mathcal{G}^{(1,l)} \times_{(3,l)} \mathcal{G}^{(2,l)}\right)_{i_l, j_l, \sigma, i_{l+1}, j_{l+1}'} = [\![\delta^1(i_l, \sigma) = i_{l+1}]\!] \cdot [\![\delta^2(j_l, \sigma) = j_{l+1}]\!]$$

$$= [\![(\delta^1 \times \delta^2)((i_l, j_l), \sigma) = (i_{l+1}, j_{l+1})]\!]$$

where $\delta^1$, $\delta^2$, $\delta^1 \times \delta^2$ refer to the transition map of $\mathcal{A}^{(1)}, \mathcal{A}^{(2)}$ and $\mathcal{A}^{(1)} \times \mathcal{A}^{(2)}$, respectively.

We have:

$$\left(\Delta^{(3)} \times_{(2,1)} \mathcal{G}^{(1,l)} \times_{(3,l)} \mathcal{G}^{(2,l)}\right)_{i_l, i_{l+1}, \sigma, j_l, j_{l+1}} = \sum_{(\sigma', \sigma'') \in \Sigma^2} \Delta_{\sigma, \sigma', \sigma''}^{(3)} \cdot \mathcal{G}_{i_l, \sigma', i_{l+1}}^{(1,l)} \cdot \mathcal{G}_{j_l, \sigma', j_{l+1}}^{(2,l)}$$

$$= \mathcal{G}_{i_l, \sigma, i_{l+1}}^{(1,l)} \cdot \mathcal{G}_{j_l, \sigma, j_{l+1}}^{(2,l)}$$

$$= [\![\delta^1(i_l, \sigma) = i_{l+1}]\!] \cdot [\![\delta^2(j_l, \sigma) = j_{l+1}]\!]$$

$\square$

Now, we are ready to prove lemma 1:

*Proof. (Lemma 1)* Proposition 1 shows that one can construct a TT equivalent to the product of K LDFAs $\{\mathcal{A}^{(k)}\}_{k \in [K]}$ whose tensor core at layer $l$ are parametrized as:

$$\Delta^{(K)} \times_{(2,1)} \mathcal{G}^{(1,l)} \times_{(3,1)} \cdots \cdots \times_{(K+1,1)} \mathcal{G}^{(K+1,l)}$$

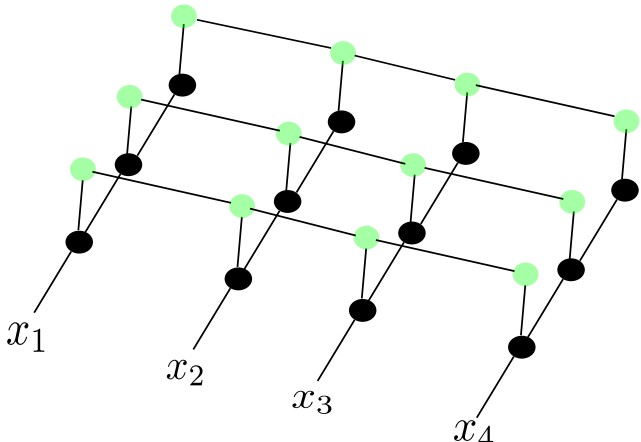

Figure 7: The structure of a TN equivalent to the product operation of 3 LDFAs (whose TT representations are depicted in green). The number of layers is equal to $L = 4$. The tensors in the bottom (in black) correspond to the TT representation of the copy tensor.

A naive construction of the tensor $\Delta^{(K)}$ scales exponentially with respect to $K$. However, constructing it using TT format (as discussed in the previous section) can be done in $\mathcal{O}(\texttt{poly}(|\Sigma|, K))$ running time.

Using the TT representation of copy tensors, the resulting TN that computes the product LDFA $\mathcal{A}^{(1)} \times \ldots \times \mathcal{A}^{(K)}$ has a 2-dimensional grid structure whose width is equal to the number of layers $L$ and whose height is equal to $K$. This TN structure can be constructed in polynomial time with respect to the size of the LDFAs, $K$, $|\Sigma|$, and $L$. Figure 7 provides a graphical illustration of this construction. $\qquad\square$

## C  Computing Marginal SHAP for General TNs

This section presents the proof of correctness of the framework introduced in section 4 to compute exactly Marginal SHAP scores of TNs with arbitrary structures. Specifically, we prove Proposition 1, Lemma 1, and Lemma 2. This section is organized into three parts, each devoted to one of these results.

### C.1  Proof of Proposition 1

Recall the statement of Proposition 1:

**Proposition.** *Define the* modified Weighted Coalitional tensor $\tilde{\mathcal{W}} \in \mathbb{R}^{n_{in} \times 2^{\otimes n_{in}}}$ *such that* $\forall (i, s_1, \ldots, s_{n_{in}}) \in [n_{in}] \times [2]^{\otimes n_{in}}$ *it holds that* $\tilde{\mathcal{W}}_{i,s_1,\ldots,s_{n_{in}}} \overset{\text{def}}{=} -W(s - 1_{n_{in}})$ *if* $s_i = 1$, *and* $\tilde{\mathcal{W}}_{i,s_1,\ldots,s_{n_{in}}} \overset{\text{def}}{=} W(s - 1_{n_{in}})$ *otherwise. Moreover, define the* marginal value tensor $\mathcal{V}^{(M,P)} \in \mathbb{R}^{\mathbb{D} \times 2^{\otimes n_{in}} \times n_{out}}$, *such that* $\forall (x, s) \in \mathbb{D} \times [2]^{\otimes n}$ *it holds that* $\mathcal{V}^{(M,P)}_{x,s,:} \overset{\text{def}}{=} V_M(x, s; P)$. *Then we have that:*

$$\mathcal{T}^{(M,P)} = \tilde{\mathcal{W}} \times_S \mathcal{V}^{(M,P)} \tag{12}$$

*where* $S \overset{\text{def}}{=} \Big\{ (k+1, k+n_{in}+1) : k \in [n_{in}] \Big\}$.

*Proof.* We first define the $\texttt{swap}(.,.)$ operation. For $n \in \mathbb{N}$, a binary vector $s \in \{0,1\}^n$ and $i \in [n]$, $\texttt{swap}(s, i)$ returns a binary vector of dimension $n$ equal to $s$ everywhere except for the $i$-th element where it is equal to 1.

We rewrite the vector $\phi_i(M, x, P)$ (Equation (1)) as follows:

$$\mathcal{T}^{M,P}_{x,i,:} = \phi_i(M, x, P) = \sum_{\substack{s \in \{0,1\}^{n_{in}} \\ s_i = 0}} W(s)\left[V_M(x, \mathtt{swap}(s, i); P) - V_M(x, s; P)\right]$$

$$= \sum_{\substack{s \in \{0,1\}^{n_{in}} \\ s_i = 1}} W(s) \cdot V_M(x, s; P) - \sum_{\substack{s \in \{0,1\}^{n_{in}} \\ s_i = 0}} W(s) \cdot V_M(x, s; P)$$

$$= \sum_{s \in \{0,1\}^{n_{in}}} \tilde{W}(s, i) \cdot V_M(x, s; P) \tag{13}$$

where: $\tilde{W}(s, i) \stackrel{\text{def}}{=} \begin{cases} W(s) & \text{if } s_i = 1 \text{ (Feature i to explain is part of the coalition)} \\ -W(s) & \text{otherwise (Feature i to explain is not part of the coalition)} \end{cases}$.

A contraction of the modified Weighted Coalitional Tensor $\tilde{\mathcal{W}}$ with the Marginal Value Tensor $\mathcal{V}^{(M,P)}$ across the dimensions defined by the set $S$ (defined in the proposition statement) straightforwardly yields Equation (12). $\qquad\square$

### C.2   The construction of the modified Weighted Coalitional Tensor $\tilde{\mathcal{W}}$ (Lemma 1)

Recall the statement of Lemma 1:

**Lemma.** *The modified Weighted Coalitional Tensor $\tilde{\mathcal{W}}$ admits a TT representation: $[\![\mathcal{G}^{(1)}, \ldots, \mathcal{G}^{(n_{in})}]\!]$, where $\mathcal{G}^{(1)} \in \mathbb{R}^{n_{in} \times 2 \times n_{in}^2}$, $\mathcal{G}^{(i)} \in \mathbb{R}^{n_i^2 \times 2 \times n_i^2}$ for any $i \in [2, n_{in} - 1]$, and $\mathcal{G}^{(n_{in})} \in \mathbb{R}^{n_{in}^2 \times 2}$. Moreover, this TT representation is constructible in $O(\log(n_{in}))$ time using $O\left(n_{in}^3\right)$ parallel processors.*

The definition of the Marginal SHAP Coalitional Tensor is:

$$\forall (i, \tilde{s}_1, \ldots, \tilde{s}_{n_{in}}) \in [n_{in}] \times [2]^{\otimes n_{in}} : \quad \tilde{\mathcal{W}}_{i, \tilde{s}_1, \ldots, s_{n_{in}}} \stackrel{\text{def}}{=} \begin{cases} -W(\tilde{s} - 1_{n_{in}}) & \text{if } \tilde{s}_i = 1 \\ W(\tilde{s} - 1_{n_{in}}) & \text{otherwise} \end{cases} \tag{14}$$

**Remark.** *Note the existence of a coordinate shift from the boolean representation that lies in $\{0, 1\}$ to TN representation which lies in $[2]$, hence the subtraction by the vector $1_{n_{in}}$. To distinguish between both representations, we use the notation $\tilde{s}$ in the sequel to refer to the TN representation of boolean values.*

In the remainder of this section, we shall provide the exact construction of the tensor cores $\{\mathcal{G}_l\}_{l \in [n_{in}]}$ implicitly mentioned in the lemma statement, followed by a proof of its correctness and its associated complexity. Before that, we introduce some notation.

**Notation.** For a vector $\tilde{s} \in [2]^n$, we denote $|\tilde{s}|$ the number of its elements equal to 1, and $\tilde{s}_{1:k} \in \mathbb{R}^k$ the vector composed of its first $k$ elements. We define the function $\Delta$ that takes as input a vector $\tilde{s} \in [2]^n$ and an integer $i \in \mathbb{N}$, and outputs 1 if $[(\tilde{s}_i = 1) \vee (n \leq i - 1)]$, $-1$ otherwise. The function $\Delta(., .)$ can be alternatively defined using the recursive formula:

$$\forall \sigma \in [2] : \Delta(\tilde{s}\sigma, i) \stackrel{\text{def}}{=} \begin{cases} -\Delta(\tilde{s}, i) & \text{if } |\tilde{s}| = i - 1 \wedge \sigma = 1 \\ \Delta(\tilde{s}, i) & \text{otherwise} \end{cases}$$

where $\Delta(\epsilon, i) = 1$ ($\epsilon$ refers to the trivial vector of dimension 0)

**High-Level Steps of the Construction.**   The general idea of the construction is to build a TT that simulates the computation of the modified Weighted Coalitional Tensor by processing its input $(i, \tilde{s}_1, \tilde{s}_2, \ldots, \tilde{s}_{n_{in}})$ from left to right in a similar fashion to state machines. The introduction of each new element $\tilde{s}_j$ in the sequence updates the state of the computation by adjusting the weight corresponding to the size of the coalition of features seen so far and retaining the relevant information required for the processing of subsequent elements.

Intuitively, two pieces of information need to be kept throughout the computation performed by the state machine:

- *The feature to explain $i$:* This information is provided at the beginning of the computation by the first leg of the tensor $\tilde{\mathcal{W}}$ and will be stored throughout the computation, using a simple copy operation. This information is needed to flip the sign when the position $i$ in the sequence is reached (Equation 14).

- *The size of the coalition $k$.* Elements of the modified Weighted Coalitional Tensor depend on the size of the coalition. This information needs to be updated throughout the processing of the input sequence. Assume at step $i \in [n_{in}]$, the size of the coalition formed from the first $i$ features is equal to $k$:

    - If $\tilde{s}_i = 1$ (feature $i$ is not part of the coalition), we transition to the state $k$ to the next step, and we normalize the weight of the coalition accordingly,
    - If $\tilde{s}_i = 2$ (feature $i$ is part of the coalition), we transition to the state $k + 1$, and normalize the weight of the coalition accordingly.

Based on this description, we propose a TT representation of $\tilde{W}$ that maintains an internal state encoded as a matrix $G^{(j)} \in \mathbb{R}^{n_{in} \times n_{in}}$. The semantics of the element $G^{(j)}_{i,k}$ in the matrix reflect the description described above.

**Construction of the core tensors $\{\mathcal{G}^{(i)}\}_{i \in [n_{in}]}$.** For simplicity, we assume that the core tensors are of order 5, so that the internal state encoding takes a matrix format and holds the semantics outlined in the above description. The transition to core tensors of order 3 as in the TT format can be performed through a suitable reshape operation. The parameterization of the core tensors $\{\mathcal{G}^{(i)}\}_{i \in [n_{in}]}$ is given as follows:

- The core tensor $\mathcal{G}^{(1)}$:

    1. If $\tilde{s_1} = 1$ (feature 1 is not part of the coalition):

    $$\mathcal{G}^{(1)}_{i,1,i',k'} = \begin{cases} (-1)^{[\![i=1]\!]} \cdot (n_{in} - 1)! & \text{if } i' = i \wedge k' = 1 \\ 0 & \text{otherwise} \end{cases}$$

    2. If $\tilde{s}_j = 2$ (feature 1 is part of the coalition):

    $$\mathcal{G}^{(1)}_{i,2,i',k'} = \begin{cases} (n_{in} - 2)! & \text{if } i' = i \wedge k' = 2 \\ 0 & \text{otherwise} \end{cases}$$

- The core tensors $\{\mathcal{G}^{(j)}\}_{j \in [n_{in}] \setminus \{1\}}$:

    1. If $\tilde{s}_j = 1$ (feature $j$ is not part of the coalition):

    $$\mathcal{G}^{(j)}_{i,k,1,i',k'} = \begin{cases} \frac{(-1)^{[\![j=i]\!]}}{j} & \text{if } i' = i \wedge k' = k \\ 0 & \text{otherwise} \end{cases} \tag{15}$$

    2. If $\tilde{s}_j = 2$ (feature $j$ is part of the coalition):

    $$\mathcal{G}^{(j)}_{i,k,2,i',k'} = \begin{cases} \frac{(k+1)}{j \cdot (n_{in} - k - 1)} & \text{if } i' = i \wedge k' = k + 1 \\ 0 & \text{otherwise} \end{cases} \tag{16}$$

The rightmost core tensor of the TT is obtained by contracting $\mathcal{G}^{(n_{in})}$ with the ones matrix $1_{n_{in} \times n_{in}}$ (The matrix whose all elements are equal to 1) at the pair of leg indices $(4, 1)$ and $(5, 2)$.

**Correctness.** The proposed TT construction is designed in such a way that it captures the dynamics outlined in the description above. The state machine maintains a sparse matrix equal to zero everywhere except for the element $(i, k)$ where $i$ corresponds to the feature to explain, and $k$ is the number of already processed features in the coalition. The following proposition formalizes this fact:

**Proposition 2.** *Fix an integer $n_{in} \geq 1$. For any $j \in [n_{in}], i \in [n_{in}], (l,k) \in [n_{in}]^2$ and $s = (s_1, \cdots, s_j) \in [2]^j$, we have:*

$$\left( \mathcal{G}^{(1)} \times_{(3,1)} \mathcal{G}^{(2)} \times \cdots \times_{(3,1)} \mathcal{G}^{(j)} \right)_{i,\tilde{s}_1,\ldots,\tilde{s}_j,:,:} = \begin{bmatrix} 0 & \cdots & 0 & \cdots & 0 \\ \vdots & \ddots & \vdots & & \vdots \\ 0 & \cdots & G_{i,|\tilde{s}_{1:j}|} & \cdots & 0 \\ \vdots & & \vdots & \ddots & \vdots \\ 0 & \cdots & 0 & \cdots & 0 \end{bmatrix} \in \mathbb{R}^{n_{in} \times n_{in}}$$

(17)

*where:*

$$G_{i,|\tilde{s}_{1:j}|} = \begin{cases} (-1)^{\Delta(\tilde{s}_{1:j},i)} \cdot \frac{|\tilde{s}_{1:j}|!(n_{in}-|\tilde{s}_{1:j}|-1)!}{j!} & \text{if } l = i \wedge |\tilde{s}_{1:j}| = k \\ 0 & \text{otherwise} \end{cases}$$

*Proof.* Fix $n_{in} \geq 1$. The proof proceeds by induction on $j$.

*Base Case.* Assume $j = 1$. Let $i \in [n_{in}]$, $(l,k) \in [n_{in}]^2$ and $\tilde{s}_1 \in [2]$. Equation (17) holds by construction of $\mathcal{G}^{(1)}$.

*General Case.* Assume Equation (17) holds for $j \in [n_{in} - 1]$, we need to show that it's also valid for $j + 1$.

Let $(l,k) \in [n_{in}]^2$ and $(\tilde{s}_1, \ldots, \tilde{s}_{j+1}) \in [2]^{j+1}$. For better readability, we adopt the following notation in the sequel:

$$\mathcal{H}^{(j)} \overset{\text{def}}{=} \mathcal{G}^{(1)} \times_{(3,1)} \mathcal{G}^{(2)} \times \cdots \times_{(3,1)} \mathcal{G}^{(j)}$$

By the induction assumption, we have:

$$\mathcal{H}^{(j+1)}_{i,\tilde{s}_1,\ldots,\tilde{s}_{j+1},l,k} = \sum_{l',k'} \mathcal{H}^{(j)}_{i,\tilde{s}_1,\ldots,\tilde{s}_j,l',k'} \cdot \mathcal{G}^{(j)}_{l',k',\tilde{s}_{j+1},l,k}$$

$$= (-1)^{\Delta(\tilde{s}_{1:j},i)} \cdot \frac{|\tilde{s}_{i:j}|! \cdot (n_{in} - |\tilde{s}_{1:j}| - 1)!}{j!} \cdot \mathcal{G}^{(j)}_{i,|\tilde{s}_{1:j}|,\tilde{s}_{j+1},l,k}$$

where the second equality is obtained by the induction assumption.

We first handle the case when $\mathcal{H}^{(j+1)}_{\tilde{s}_1,\ldots,\tilde{s}_{j+1},l,k} = 0$.

By construction of $\mathcal{G}^{(j+1)}$, we have:

$$(l \neq i) \vee (k \neq |\tilde{s}_{1:j+1}|) \implies \mathcal{G}^{(j+1)}_{i,|\tilde{s}_{1:j}|,1,l,k} = 0$$

$$\implies \mathcal{H}^{(j+1)}_{i,\tilde{s}_1,\ldots,\tilde{s}_j,1,l,k} = 0$$

Next, we analyze the case: $l = i \wedge |s_{1:j+1}| = k$. We split into two cases:

*Case $s_{j+1} = 1$ (The feature $j+1$ is not part of the coalition):* In this case, note that: $|s_{1:j+1}| = |s_{1:j}|$.

We have:

$$\mathcal{H}^{(j+1)}_{i,s_1,\ldots,s_{j+1},i,|s_{1:j+1}|} = (-1)^{\Delta(s_{1:j},i)} \cdot \frac{|s_{i:j}|! \cdot (n_{in} - |s_{1:j}| - 1)!}{j!} \cdot \frac{(-1)^{[i=j+1]}}{j}$$

$$= (-1)^{\Delta(s_{1:j+1},i)} \cdot \frac{|s_{i:j+1}|! \cdot (n_{in} - |s_{1:j+1}| - 1)!}{(j+1)!}$$

*Case $s_{j+1} = 2$ (The feature $j+1$ is not part of the coalition).* In this case, note that: $|s_{1:j+1}| = |s_{1:j}| + 1$, and $\Delta(s_{1:j+1},i) = \Delta(s_{1:j},i)$.

We have:

$$\mathcal{H}^{(j+1)}_{i,s_1,\ldots,s_{j+1},i,|s_{1:j+1}|} = (-1)^{\Delta(s_{1:j+1},i)} \cdot \frac{|s_{1:j}|! \cdot (n_{in} - |s_{1:j+1}|)!}{j!} \cdot \frac{|s_{1:j+1}|}{(j+1) \cdot (n_{in} - |s_{1:j+1}|)}$$

$$= (-1)^{\Delta(s_{1:j+1},i)} \cdot \frac{|s_{1:j+1}|! \cdot (n_{in} - |s_{1:j+1}| - 1)!}{(j+1)!}$$

$\square$

**Complexity.** The core tensors $\{\mathcal{G}^{(j)}\}_{j\in[n_{in}]}$ are extremely sparse. By leveraging this sparsity property, each core tensor can be constructed in $\mathcal{O}(1)$ time using $\mathcal{O}(n_{in}^2)$ parallel processors. By parallelizing the construction of all core tensors, this leads to a number of $\mathcal{O}(n_{in}^3)$ parallel processors. Yet, it should be observed that the construction of the leftmost tensor $\mathcal{G}^{(1)}$ requires the computation of the factorial terms $(n_{in}-1)!$ and $(n_{in}-2)!$. This operation can be performed using a parallel scan strategy using $\mathcal{O}(n_{in})$ parallel processors with $\mathcal{O}(\log(n_{in}))$ running time.

To summarize, the total computation of the TT corresponding to the modified Weighted Coalitional Tensor requires $\mathcal{O}(n_{in}^3)$ parallel processors and runs in $\mathcal{O}(\log(n_{in}))$ time.

### C.3 The construction of the Marginal SHAP Value Tensor $\mathcal{V}^{(M,P)}$ (Lemma 2)

The objective of this section is to prove the Lemma 2. Recall the statement of this lemma:

**Lemma.** *Let $\mathcal{T}^M$ and $\mathcal{T}^P$ be two TNs implementing the model $M$ and the probability distribution $P$, respectively. Then, the marginal value tensor $\mathcal{V}^{(M,P)}$ can be computed as:*

$$\left[\mathcal{M}^{(1)} \circ \ldots \circ \mathcal{M}^{(n_{in})}\right] \times_{S_1} \mathcal{T}^M \times_{S_2} \mathcal{T}^P$$

*where:*

- *$S_1$ and $S_2$ are instantiated such that for all $k \in \{1,2\}$ it holds that: $S_k \stackrel{\text{def}}{=} \{(5-k)\cdot i, i) : i \in [n_{in}]\}$,*

- *For any $i \in [n_{in}]$, $(\sigma, s, \sigma', \sigma'') \in [N_i] \times [2] \times [N_i]^{\otimes 2}$, we have:*

$$\mathcal{M}^{(i)}_{\sigma,s,\sigma',\sigma''} = \begin{cases} 1 & \text{if } (\sigma'' = \sigma \wedge s = 1) \vee (\sigma'' = \sigma' \wedge s = 2) \\ 0 & \text{otherwise} \end{cases} \tag{18}$$

Before providing the proof of Lemma 2, we first introduce an operator which will be useful for the proof:

**The** do **operator [82].** Let $S$ be a finite set. The do operator takes as input a triplet $(\sigma, s, \sigma') \in S \times [2] \times S$ and returns $\sigma$ if $s = 1$, $\sigma'$ otherwise. The collection of Tensors $\mathcal{M}^{(i)}$ parametrized as in Equation (18) can be seen as *tensorized* representations of the do operator [82]. When $S = [N_i]$, we have, for any $(\sigma, s, \sigma', \sigma'') \in [N_i] \times [2] \times [N_i] \times [N_i]$, $\mathcal{M}^{(i)}_{\sigma,\sigma',\sigma''} = 1$ if and only if $\sigma'' = \boldsymbol{do}(\sigma, s, \sigma')$. By induction, for any $(x, s, x', x'') \in \mathbb{D} \times [2] \times \mathbb{D}^{\otimes 2}$, we have:

$$\left(\mathcal{M}^{(1)} \circ \ldots \circ \mathcal{M}^{(n_{in})}\right)_{x,s,x',x''} = 1 \iff \forall i \in [n_{in}]: \ x_i'' = \boldsymbol{do}(x_i, s_i, x_i', x_i'') \tag{19}$$

Now, we are ready to provide the proof of Lemma 2:

*Proof. (Lemma 2)* Let $(x, s, y) \in \mathbb{D} \times [2]^{\otimes n_{in}} \times [n_{out}]$, we have:

$$\mathcal{V}^{(M,P)}_{x,s,y} = \mathbb{E}_{x' \sim \mathcal{T}^P} \left[ \mathcal{T}^M_{\boldsymbol{do}(x_1', s_1, x_1), \ldots, \boldsymbol{do}(x_{n_{in}}', s_{n_{in}}, x_{n_{in}}), y} \right]$$

$$= \mathbb{E}_{x' \sim \mathcal{T}^P} \left[ \sum_{x'' \in \mathbb{D}} \left(\mathcal{M}^{(1)} \circ \ldots \circ \mathcal{M}^{(n_{in})}\right)_{x,s,x',x''} \cdot \mathcal{T}^M_{x'',y} \right]$$

$$= \mathbb{E}_{x' \sim \mathcal{T}^P} \left[ \left(\mathcal{M}^{(1)} \circ \ldots \circ \mathcal{M}^{n_{in}}\right) \times_{S_1} \mathcal{T}^M \right]_{x,s,x',y}$$

$$= \sum_{x' \in \mathbb{D}} \mathcal{T}^P_{x'} \cdot \left[ \left(\mathcal{M}^{(1)} \circ \ldots \circ \mathcal{M}^{n_{in}}\right) \times_{S_1} \mathcal{T}^M \right]_{x,s,x',y}$$

$$= \left[ \left(\mathcal{M}^{(1)} \circ \ldots \circ \mathcal{M}^{n_{in}}\right) \times_{S_1} \mathcal{T}^M \times_{S_2} \mathcal{T}^P \right]_{x,s,y}$$

$\square$

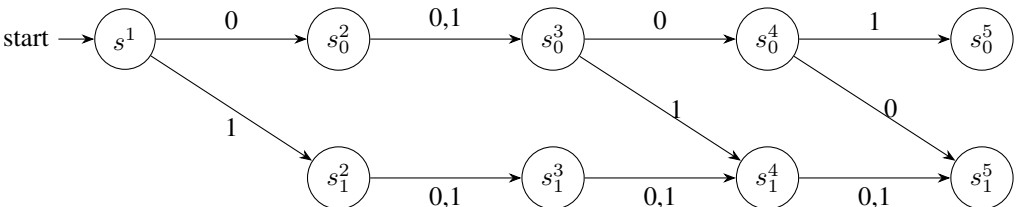

Figure 8: An LDFA that accepts the set of satisfying assignments of the disjunctive clause: $C = x_1 \vee \neg x_3 \vee x_4$. The lowest states in the grid ($\{s_1^2, s_1^3, s_1^4, s_1^5\}$ tracks the satisfiability of the running assignment. Satisfying assignments are those that reach the state $s_1^5$

## D    Computing Marginal SHAP for Tensor Trains

This section contains the proof of the #P-Hardness of computing SHAP for General TNs (Proposition 2), as well as a proof of the fact that SHAP for TTs is in NC.

### D.1    SHAP for general TNs is #P-Hard

Proposition 2 states the following:

**Proposition.** *Computing Marginal SHAP scores for general TNs is #P-HARD.*

*Proof.* The proof builds on known connections between model counting and the computation of SHAP values. Recall that, given a class of classifiers $\mathcal{C}$, model counting refers to the problem of determining the number of inputs classified as positive by a classifier in $\mathcal{C}$. The following classical lemma established by both [106] and [6]:

**Lemma 2.** *Given a class of classifiers $\mathcal{C}$, the model counting problem for $\mathcal{C}$ is polynomial-time reducible to the problem of computing Marginal SHAP values for the class $\mathcal{C}$ under the uniform distribution.*

We note that although the original lemma addresses a different value function, specifically the Conditional SHAP variant based on conditional expectations — the Marginal SHAP and Conditional SHAP formulations coincide under the feature independence assumption [103]. Consequently, the hardness result established in that setting also applies to the general Marginal SHAP case.

Now, to prove Proposition 2 using Lemma 2, we use the class of CNF formulas as a proxy. It is a classical result in complexity theory that the model counting problem for CNFs is #P-Complete [7]. Our goal is to reduce this problem to the model counting problem for TNs:

**Lemma 3.** *There exists a polynomial-time algorithm that takes as input an arbitrary CNF formula $\Phi$ and produces a TN that computes an equivalent Boolean function.*

*Proof.* Let $\Phi$ be a CNF formula composed of $L$ clauses. The reduction proceeds as follows: For each clause, construct an equivalent Layered Deterministic Finite Automaton (LDFA). This transformation can be performed in linear time with respect to the number of variables in the clause (see Figure 8 for an illustrative example). Then, construct the TN that corresponds to the product of the resulting LDFAs. This TN simulates the conjunction over all clauses of the formula $\Phi$. The polynomial time complexity of this procedure is guaranteed by Lemma 1. $\square$

The proof of Lemma 3, combined with our earlier claims, completes the proof of Proposition 2:

*Proof.* (*Proposition 2*) By Lemma 2, the model counting problem for CNFs is polynomially reducible to computing Marginal SHAP values under the uniform distribution. Lemma 3 ensures that any CNF formula is reducible in polynomial time into an equivalent TN. Moreover, the uniform input distribution can be encoded as a product of rank-one tensors in linear time with respect to the number of input variables; for instance, by the tensor $\frac{1}{2}(e_1^1 + e_2^1) \circ \ldots \circ \frac{1}{2}(e_1^n + e_2^n)$. This completes the reduction and proves the #P-Hardness of computing Marginal SHAP scores for general TNs. $\square$

## D.2 Marginal SHAP for TTs is in $\text{NC}^2$.

In this section, we provide the details of the algorithmic construction to compute the exact marginal SHAP value for TTs in poly-logarithmic time using a polynomial number of parallel processors.

The algorithmic construction we propose stems its correctness from Theorem 1 which states the following:

**Theorem.** *Let $\mathcal{T}^M = [\![\mathcal{I}^{(1)}, \ldots, \mathcal{I}^{(n_{in})}]\!]$ and $\mathcal{T}^P = [\![\mathcal{P}^{(1)}, \ldots, \mathcal{P}^{(n_{in})}]\!]$ be two TTs corresponding to the model to interpret and the data-generating distribution, respectively. Then, the Marginal SHAP Tensor $\mathcal{T}^{(M,P)}$ can be represented by a TT parametrized as:*

$$[\![\mathcal{M}^{(1)} \times_{(4,2)} \mathcal{I}^{(1)} \times_{(3,2)} \mathcal{P}^{(1)} \times_{(2,2)} \mathcal{G}^{(1)}, \ldots \ldots, \mathcal{M}^{(n_{in})} \times_{(4,2)} \mathcal{I}^{(n_{in})} \times_{(3,2)} \mathcal{P}^{(n_{in})} \times_{(2,2)} \mathcal{G}^{(n_{in})}]\!] \tag{20}$$

*where the collection of Tensors $\{\mathcal{G}^{(i)}\}_{i \in [n_{in}]}$ and $\{\mathcal{M}^{(i)}\}_{i \in [n_{in}]}$ are implicitly defined in Lemma 1 and Lemma 2, respectively.*

Theorem 1 is a corollary of Proposition 1: Replace $\tilde{\mathcal{W}}$ with its TT parametrization (Lemma 1), and replace $\mathcal{T}^M$ and $\mathcal{T}^P$ with their corresponding TT parametrizations in the formulation of the Marginal Value Tensor in Lemma 2.

Next, our focus shall be placed on the computational aspect of computing Marginal SHAP scores for TTs by leveraging the result of Theorem 1 to show that this problem is in NC.

Denote:

- $\mathcal{T}^M = [\![\mathcal{I}^{(1)}, \ldots, \mathcal{I}^{(n_{in})}]\!]$ a TT model such that, for each $i \in [n_{in}]$, $\mathcal{I}^{(i)}$ is in $\mathbb{R}^{d_{M,i} \times N_i \times d_{M,i+1}}$ ($d_{M,1} = 1$ and $d_{M,n_{in}+1} = n_{out}$),

- $\mathcal{T}^P = [\![\mathcal{P}^{(1)}, \ldots, \mathcal{P}^{(n_{in})}]\!]$ a TT implementing a probability distribution over $\mathbb{D}$ such that, for each $i \in [n_{in}]$, $\mathcal{P}^{(i)}$ is in $\mathbb{R}^{d_{P,i} \times N_i \times d_{P,i+1}}$ ($d_{P,1} = d_{P,n_{in}+1} = 1$),

- An input instance $x = (x_1, \ldots, x_{n_{in}}) \in \mathbb{D}$,

In light of Theorem 1, a typical parallel scan procedure to compute the matrix $\Phi(\mathcal{T}^M, x, \mathcal{T}^P)$ runs as follows:

- **Level 0.** Compute in parallel the following $n_{in}$ tensors. For $i \in [n_{in}]$:

$$\mathcal{H}_0^{(i)} \overset{\text{def}}{=} \mathcal{M}^{(i)} \times_{(4,2)} \mathcal{I}^{(i)} \times_{(3,2)} \mathcal{P}^{(i)} \times_{(2,2)} \mathcal{G}^{(i)} \times_{(1,1)} e_{x_i}^{N_i} \in \mathbb{R}^{\left(n_{in}^2 \times n_{d_{P,i}} \times n_{d_{M,i}}\right)^{\otimes 2}} \tag{21}$$

- **Level 1 to $\log(\mathbf{n_{in}})$.** At step $j \in [\lfloor \log(n_{in}) \rfloor]$, perform a contraction operation over Neighboring Tensors. For $i \in [\frac{n_{in}}{2^j}]$:

$$\mathcal{H}_j^{(i)} = \mathcal{H}_{j-1}^{2 \cdot (i-1)} \times_S \mathcal{H}_{j-1}^{2 \cdot i} \tag{22}$$

where $S \overset{\text{def}}{=} \{(7-k, k) : k \in [3]\}$.

By Theorem 1, the produced matrix at the last step is equal to $\Phi(\mathcal{T}^M, x, \mathcal{T}^P)$.

**Complexity.** Each tensor contraction operation in Equations (21) and (22) requires at most $\mathcal{O}\left(n_{in}^4 \cdot \max_{i \in [n_{in}]} d_M \cdot \max_{i \in [n_{in}]} d_{P,i}\right)$. At level 0, we need to perform in parallel $n_{in}$ operations of this kind upgrading the number of required parallel processors to $\mathcal{O}\left(n_{in}^5 \cdot \max_{i \in [n_{in}]} d_{M,i} \cdot \max_{i \in [n_{in}]} d_{P,i}\right)$. As for the running time, the depth of the circuit performing the tensor contractions is bounded by $\prime\left(\log(n_{in}) + \log(\max_{i \in [n_{in}]} d_{M,i}) + \log(\max_{i \in [n_{in}]} d_{P,i})\right)$. The depth of the total circuit to compute Marginal SHAP for TTs is thus bounded by $\mathcal{O}\left(\log(n_{in}) \cdot [\log(n_{in}) + \log(\max_{i \in [n_{in}]} d_{M,i}) + \log(\max_{i \in [n_{in}]} d_{P,i})]\right)$.

# E  Tightening Complexity Results for Other ML Models

In this section, we employ NC reductions to show that the problem of computing Marginal SHAP scores for certain popular model classes (Linear RNNs, decision trees, ensemble trees, and linear models) lies also in $\text{NC}^2$ (Theorem 2).

An NC reduction is a type of many-one reduction where the transformation function from instances of problem $A$ to instances of problem $B$ is computable by a uniform family of Boolean circuits with polynomial size and polylogarithmic depth. This ensures that the reduction itself can be performed efficiently in parallel. A crucial property of the class NC is its closure under NC reductions: if a problem $A$ is NC-reducible to a problem $B$, and $B$ is in NC, then $A$ is also in NC [7].

## E.1  Reduction from Linear RNNs and Linear Models to TTs.

We assume without loss of generality that $N = N_1 = N_2 = \ldots = N_{n_{in}}$. This assumption reflects the practical use of RNNs, such as in Natural Language Processing (NLP) applications, where elements of the sequence are assumed to belong to a finite vocabulary.

**Linear RNNs and TTs.**  General TTs can be seen as non-stationary generalizations of Linear RNNs at a fixed window. Interestingly, Rabusseau *et al.* [94] showed that stationary TTs are strictly equivalent to second-order linear RNNs (see section B for a formal definition of second-order linear RNNs). Indeed, reformulating the equations governing the dynamics of a second-order linear RNNs in a *tensorized* format, it can be shown using careful algebraic calculation that:

$$\begin{bmatrix} \mathbf{h}_t \\ 1 \end{bmatrix} = \left( \tilde{\mathcal{T}} \times_{(2,1)} I \right) \times_{(1,1)} \begin{bmatrix} \mathbf{h}_{t-1} \\ 1 \end{bmatrix} \times_{(2,1)} e_{x_t}^N \tag{23}$$

Where it holds that:

- $I = \begin{bmatrix} I_{N \times N} \\ 1_N^T \end{bmatrix} \in \mathbb{R}^{(N+1) \times N}$

- $\mathcal{T} \in \mathbb{R}^{(d+1) \times (N+1) \times (d+1)}$ is such that:

  1. If $i \in [d] \wedge j \in [N]$:
  $$\tilde{\mathcal{T}}_{i,j,:} = \begin{bmatrix} \mathcal{T}_{i,j,:} \\ 0 \end{bmatrix}$$

  2. If $i = d + 1 \wedge j \in [N]$:
  $$\tilde{\mathcal{T}}_{i,j,:} = \begin{bmatrix} W_{j,:} \\ 0 \end{bmatrix}$$

  3. If $i \in [d] \wedge j = N + 1$:
  $$\tilde{\mathcal{T}}_{i,j,:} = \begin{bmatrix} U_{j,:} \\ 0 \end{bmatrix}$$

All the other elements are set to $0$.

The additional dummy neuron at dimension $d + 1$ (whose value is always equal to 1) is added to the model to account for additive terms present in the linear RNN dynamics equation.

Consequently, a second-order linear RNN $R$ at a bounded window $n_{in}$ is equivalent to the stationary TT $\mathcal{T}^R$ parametrized as:

$$\left[ \left[ \left( \tilde{\mathcal{T}} \times_{(2,1)} I \right) \times_{(1,1)} \begin{bmatrix} h_0 \\ 1 \end{bmatrix}, \tilde{\mathcal{T}} \times_{(2,1)} \tilde{W} \ldots, \left( \tilde{\mathcal{T}} \times_{(2,1)} \tilde{W} \right) \times_{(3,1)} \begin{bmatrix} O \\ 0_{N+1} \end{bmatrix} \right] \right] \tag{24}$$

**Reduction and Complexity.**  The SHAP computational problem for Linear RNNs under TT distributions takes as input an RNN $R$ of size $d$ mapping sequences of elements in $[N]$ to $\mathbb{R}^{n_{out}}$, an input sequence $x = (x_1, \ldots, x_{n_{in}}) \in [N]^{\otimes n_{in}}$ and $\mathcal{T}^P$ and outputs the SHAP Matrix $\Phi(R, x, \mathcal{T}^P) \in \mathbb{R}^{n_{in} \times n_{out}}$ (Equation ).

Fix an input instance $(R, x, \mathcal{T}^P)$. Our objective is to construct in poly-logarithmic time using a poly-nomial number of parallel processors an input instance of the SHAP problem for TTs: $(\mathcal{T}^R, \bar{x}, \mathcal{T}^{\bar{P}})$ such that $\Phi(R, x, \mathcal{T}^P) = \Phi(\mathcal{T}^R, \bar{x}, \mathcal{T}^{\bar{P}})$. The detailed reduction strategy of the SHAP problem of Linear RNNs to the SHAP problem of TTs is detailed as follows:

- **Reduction of x and $\mathcal{T}^P$**: The input instance $x$ and the tensor $\mathcal{T}^P$ are mapped trivially to the instance $\bar{x}$ and the tensor $\mathcal{T}^{\bar{P}}$ of the input instance of the SHAP problem for TTs. This operation runs in constant time using a linear number of parallel processors with respect to $n_{in}$ and the size of $\mathcal{T}^P$

- **Reduction of the linear RNN R**: Equation (24) hints at a straightforward strategy to reduce linear RNNs into equivalent TTs at a bounded window:

  - Apply the tensor contraction operation: $\mathcal{H} = \tilde{\mathcal{T}} \times_{(2,1)} I$. This operation runs in $\mathcal{O}(\log(N))$ using $\mathcal{O}(d^2 \cdot N)$ parallel processors
  - Compute Leftmost and Rightmost matrices:

    * Leftmost Matrix $\mathcal{H}^{(1)}$: $\mathcal{H}^{(1)} = \mathcal{H} \times_{(1,1)} \begin{bmatrix} h_0 \\ 1 \end{bmatrix}$.
      This operation runs in $\mathcal{O}(\log(d))$ using $\mathcal{O}(d^2 \cdot N)$ parallel processors.
    * Rightmost Matrix: $\mathcal{H}^{(n_{in})} = \mathcal{H} \times_{(3,1)} \begin{bmatrix} O \\ 0_{N+1} \end{bmatrix}$.
      This operation runs in $\mathcal{O}(\log(d))$ using $\mathcal{O}(d^2 \cdot N)$ parallel processors.

  - Run in parallel $\mathcal{O}(n_{in} \cdot N \cdot d^2)$ parallel workers to place in the input tape of the SHAP problem for TTs the following TT:

$$\left[\!\!\left[ \mathcal{H}^{(1)}, \underbrace{\mathcal{H}, \ldots, \mathcal{H}}_{(n_{in}-2)\text{times}}, \mathcal{H}^{(n_{in})} \right]\!\!\right]$$

The total reduction strategy runs in $\mathcal{O}(\max(\log(N), \log(d)))$ using $\mathcal{O}(n_{in} \cdot N \cdot d^2 + |\mathcal{T}^P|)$ parallel processors.

## E.2   Reduction from Tree Ensembles to TTs.

The objective of this subsection is to show that computing SHAP for Tree Ensembles is NC-reducible to the SHAP problem for TTs.

We first note the fact that if computing SHAP for DTs is efficiently parallelizable, then so is Ensemble Trees. Indeed, by the linearity property of the SHAP score, computing SHAP for Ensemble Trees is obtained as a weighted sum of SHAP scores of the trees forming the ensemble. This operation adds $\mathcal{O}(\log(N))$ depth to the circuit that computes SHAP scores of single DTs, where $N$ refers to the number of trees in the ensemble. This implies the following fact:

**Fact.** *Fix a distribution class $P$. Computing Marginal SHAP for the class of Decision Trees under $P$ is in* NC *implies computing Marginal SHAP for Ensemble Trees is also in* NC.

In light of this fact, we dedicate the rest of this section to examining the specific case of DTs and their reduction into TTs.

**Decision Trees as Disjoint DNFs.**   We first propose a representation of Tree-based models. This representation has been shown to be more amenable to parallelization and forms the core of the pre-processing step as the GPUTreeSHAP algorithm [84]. Define the following collection of predicates:

$$p_{ij} \stackrel{\text{def}}{=} [\![ x_i = j ]\!]$$

where $i \in [n_{in}]$, and $j \in [N_i]$. A DT $T$ can be equivalently represented as a *disjoint DNF* formula over the predicates $\{p_{ij}\}_{i \in [n_{in}], j \in [N_i]}$ as [81]:

$$\Phi_T = C_1 \vee C_2 \vee \ldots \vee C_L$$

where $C_L$ is a conjunctive clause.

Clauses $\{C_j\}_{j \in L}$ satisfy the disjointness property: For two different clauses, the intersection of their satisfying assignments is empty. The conversion of Decision Trees into equivalent Disjoint DNFs has been shown to run in polynomial time with respect to the number of edges of the DT.

In the sequel, we assume that DTs are represented using the Disjoint DNF formalism. In practice, the conversion of a DT into this format can be performed offline (in polynomial time), and stored as such, in the same fashion of the GPUTreeSHAP Algorithm [84].

We begin by showing how DTs can be encoded in TT format. We assume an arbitrary ordering $X_1, \ldots, X_{n_{in}}$ of input features of the DT:

**Proposition 3.** *A DT $T$ encoded as a disjoint DNF $\Phi_T = C_1 \vee \ldots \vee C_L$ is equivalent to a TT parametrized as:*

$$\left[\!\!\left[\mathcal{I}^{(1)} \times_{(1,1)} 1_L, , \mathcal{I}^{(2)}, \ldots, \mathcal{I}^{(n_{in}-1)}, \mathcal{I}^{(n_{in})} \times_{(3,1)} 1_L\right]\!\!\right] \tag{25}$$

*such that for $l \in [n_{in}]$, the tensor $\mathcal{I}^{(l)} \in \mathbb{R}^{L \times N_l \times L}$ is such that:*

$$\mathcal{I}^{(l)}_{i,j,k} = \begin{cases} 1 & \text{if } (p_{lj} \in C_k \wedge i = k) \vee \forall j \in [N_i] : p_{lj} \notin C_i \\ 0 & \text{otherwise} \end{cases}$$

*Proof.* Let $T$ be a a DT, and $\Phi_T = C_1 \vee \ldots \vee C_L$ be its equivalent representation into a disjoint DNF format. For $k \in [n_{in}]$, we shall use the notation $C_l^k$ to denote the restriction of the clause $l$ to include only predicates in $\{p_{ij}\}_{i \leq i}$.

We claim that for any $k \in [n_{in}]$, and any $x = (x_1, \ldots, x_{N_{n_{in}}}) \in \mathbb{D}$, the TT (of length $k$) parametrized as:

$$\left[\!\!\left[\mathcal{I}^{(1)} \times_{(1,3)} 1_L, \mathcal{I}^{(2)}, \ldots, \mathcal{I}^{(k)}\right]\!\!\right]$$

is such that:

$$\mathcal{I}_{x_1, x_2, \ldots, x_k, :} = \begin{bmatrix} x_{1:k} \models C_1^k \\ \vdots \\ x_{1:k} \models C_L^k \end{bmatrix}$$

If this claim holds, then for $k = n$, and by the disjointness property of $\Phi_T$, we have for any $x = (x_1, \ldots, x_{n_{in}})$:

$$f_T(x_1, \ldots, ) = \mathcal{T}^T_{x_1, \ldots, x_{n_{in}}, :} \cdot 1_L$$

which completes the proof of the proposition.

We prove this claim by induction on $k$.

Base case: For $k = 1$, fix $(x_1, l) \in [N_1] \times [L]$. By construction of $\mathcal{I}^{(1)}$, we have:

$$(\mathcal{I}^{(1)} \times_{(1,1)} 1_L)_{i_1, l} = \sum_{l' \in [L]} \mathcal{I}^{(1)}_{l', x_1, l}$$

$$= \mathcal{I}^{(1)}_{l, i_1, l} = [\![(p_{x_1 j} \in C_1) \vee \forall j \in [N_1] : (p_{1j} \in C_l) \vee \forall j \in [N_1] : p_{1j} \in C_l]\!]$$

$$= [\![x_1 \models C_l^1]\!]$$

General Case. Assume the claim holds for $k$, we show that it also holds for $k + 1$.

Let $(x_1, \ldots x_{k+1}) \in [N_1] \times \ldots \times [N_{k+1}]$ and $l \in [L]$. We have:

$$\mathcal{I}_{x_1, x_2, \ldots x_{k+1}, l} = \sum_{l' \in [L]} \mathcal{I}_{x_1, \ldots, x_k, l'} \cdot \mathcal{I}^{(l+1)}_{l', x_{k+1}, l}$$

$$= \mathcal{I}_{x_1, \ldots, x_k, l} \cdot \mathcal{I}^{(l+1)}_{l, x_{k+1}, l}$$

$$= [\![x_{1:k} \models C_l^k]\!] \cdot [\![(p_{x_{k+1} j} \in C_l)$$

$$\vee \forall j \in [N_1] : (p_{(k+1)j} \in C_l) \vee \forall j \in [k+1] : p_{(k+1)j} \in C_l]\!]$$

$$= [\![x_{1:k+1} \models C_l^{k+1}]\!]$$

$\square$

**Reduction and Complexity.** Proposition 3 suggests the following strategy to reduce a DT encoded as a Disjoint DNF into an equivalent TT.

Fix a Disjoint DNF $\Phi_T = C_1 \vee \ldots \vee C_L$. The granularity of the parallelization strategy is at the literal level: Each parallel processor is dedicated to processing a specific literal $p_{ij}$ in a clause $C_l$: If a literal $p_{ij}$ appears in the clause $C_l$, the processor sets the value $\mathcal{I}_{l,j,l}^{(i)}$ to 1. The correctness of this parallel schema is guaranteed by Proposition 3. The running time complexity of this parallel procedure is $\mathcal{O}(1)$. The number of parallel processors is $\mathcal{O}(L \cdot n_{in})$.

# F    Computing Marginal SHAP for BNNs

In this section, we present the proof of Theorem 3 which provides fine-grained parameterized complexity for the problem of computing SHAP for BNNs. Recall this theorem's statement:

**Theorem.** *Let $P$ be either the class of empirical distributions, independent distributions, or the class of TTs. We have that:*

1. ***Bounded Depth:*** *The problem of computing SHAP for BNNs under any distribution class $P$ is* PARA-NP-HARD *with respect to the network's* depth *parameter.*

2. ***Bounded Width:*** *The problem of computing SHAP for BNNs under any distribution class $P$ is in* XP *with respect to the* width *parameter.*

3. ***Bounded Width and Sparsity.*** *The problem of computing SHAP for BNNs under any distribution class $P$ is in* FPT *with respect to the* width *and* reified cardinality *parameters.*

We split this section into two parts: The first part is dedicated to proving that computing SHAP for BNNs with bounded depth remains intractable. The second part provides the details of a procedure that builds an equivalent TT to a given BNN. The complexity analysis of this procedure will yield the results of items (2) and (3) of the Theorem.

## F.1    SHAP for BNNs with constant depth is intractable

We will demonstrate that computing SHAP values for BNNs remains intractable even when the network is restricted to a depth of just one.

**Proposition 4.** *Given a BNN $B$ with one hidden layer, and some input $x$, then it holds that computing SHAP for $B$ and $x$ is #P-Hard.*

*Proof.* We prove that computing SHAP values for a Binarized Neural Network (BNN) with just a single hidden layer is already #P-Hard. Specifically, this is shown via a reduction from the counting variant of the classic 3SAT problem — namely, #3SAT — which is known to be #P-Hard [7].

We begin by referencing the result of [6], which showed that computing SHAP values for a model $f$ and input $x$, even under the simple assumption that the distribution $\mathcal{D}_p$ is *uniform*, is as hard as model counting for $f$. Consequently, if model counting for $f$ is #P-Hard, so is SHAP computation—this follows directly from the efficiency axiom under uniform distributions. Therefore, to establish #P-Hardness of SHAP for a model $f$, it suffices to reduce from a model with known #P-Hard model counting (e.g., a 3CNF formula). Since uniform distributions are a special case of independent distributions and significantly simpler than structured models like tensor trains (TTs), the hardness results extend naturally to our setting as well.

Consider a 3-CNF formula $\phi := t_1 \wedge t_2 \wedge \ldots \wedge t_m$, where each clause $t_i$ is a disjunction of three literals: $t_i := x_j \vee x_\ell \vee x_k$. We construct a Binarized Neural Network (BNN) $B$ with a single hidden layer over the input space $\{-1, 1\}^n$—a setting that already implies hardness for more general discrete domains. Each input neuron of the BNN corresponds to a variable in $\phi$, so the input layer has $n$ neurons. The hidden layer contains $m$ neurons, one for each clause in $\phi$. For a clause $t_i$, if a variable $x_j$ appears positively, we connect input neuron $j$ to hidden neuron $i$ with weight $+1$; if it appears negatively, the weight is $-1$. Each hidden neuron is assigned a bias of $\frac{3}{2}$. Finally, we add a single output neuron with a bias of $-(m - \frac{1}{2})$.

We now show that every satisfying assignment to $\phi$ yields a "True" output in $B$, and every unsatisfying assignment yields "False". A clause $t_i$ is satisfied if at least one of its variables evaluates to True. This occurs either when a non-negated variable is assigned 1 (a "double-positive" case) or when a negated variable is assigned $-1$ (a "double-negative" case). In the BNN, both cases contribute 1 to the corresponding hidden neuron, since input and weight signs align. Thus, if any variable in $t_i$ satisfies the clause, the corresponding neuron receives at least one input of 1. With a bias of $\frac{3}{2}$, even if the remaining two inputs contribute $-1$ each, the total input is $\frac{5}{2} - 2 = \frac{1}{2} > 0$, so the step function outputs 1. Therefore, satisfied clauses activate their corresponding neurons. Finally, since the output neuron has a bias of $-(m - \frac{1}{2})$, it outputs 1 if and only if all $m$ hidden neurons are active, meaning all clauses are satisfied.

Since we have shown that each satisfying assignment of the BNN $B$ corresponds to a satisfying assignment of $\phi$, it follows that the model counting problem is the same for both. Therefore, for this BNN with only a single hidden layer, and by the result of [6], computing SHAP is #P-Hard. $\qquad\square$

## F.2 Compiling BNNs into Equivalent TTs

In this section, we provide the details of the procedure of compiling BNNs into Equivalent TTs, to show that the problem of computing SHAP for BNNs in bounded width (resp. bounded width and sparsity) is in XP (resp. FPT).

### F.2.1 Warm Up: Compilation of a single-neuron BNN with Depth 1 into TT.

As an initial step, we present in this subsection a construction for representing the activation function of a single neuron in a BNN using TTs. This construction serves as a basic building block for simulating multi-layered BNN architectures by means of the TT formalism.

Let $\mathbf{w} = (w_1, \dots w_n) \in \{-1, 1\}^n$ be the weight parameters of a BNN with a single output neuron $n_{out}$. Its activation as computed using the reified cardinality representation is given as:

$$n_{out}(x_1, \dots x_n; R) = \left[\!\!\left[ \sum_{i=1}^{n} l_i \geq R \right]\!\!\right] \tag{26}$$

where: $R$ is its reified cardinality parameter, and $l_i \overset{\text{def}}{=} \begin{cases} x_i & \text{if } w_i = 1 \\ \neg x_i & \text{otherwise} \end{cases}$.

To understand how the construction of an equivalent TT to simulate this BNN operates, we will employ the LDFA formalism previously introduced in subsection B.3.

The main idea of this construction is to build an LDFA whose states function as counters, tracking the number of satisfied literals $\{l_i\}_{i \in [n]}$ (Equation (26). When this count exceeds the specified cardinality threshold $R$, the output neuron $n_{out}$ is activated; otherwise, it remains inactive. Crucially, the number of states at each layer in LDFA scales linearly with the parameter $R$.

Formally, the LDFA that simulates the neuron $n_{out}$ (Equation (26)) is given as follows:

- *Number of layers:* $n$,
- *State space:* For each layer $l \in [n]$, $S_l = \{0, R\}$
- *The initial state:* $s_0^1$ (The state at Layer 1 corresponding to $R = 0$)
- *Transition function:* For a layer $l \in [n-1]$,

$$\delta(s_r^{(l)}, \sigma) = \begin{cases} s_{r+1}^{(l+1)} & \text{if } (r \neq R) \wedge [(w_l = 1 \wedge \sigma = 1) \vee (w_l = -1 \wedge \sigma = 0)] \\ s_r^{(l+1)} & \text{otherwise} \end{cases}$$

- *The final state:* $s_R^{(n)}$

### F.2.2 Compilation of a single Layer into TT.

In this section, we will move from BNNs with a single neuron into multi-layered BNNs. We will show how one can construct a TT equivalent to a given BNN in $O(R^W \cdot \texttt{poly}(D, \max_{i \in [n_{in}]} N_i, n_{in})$,

where $R$ is the reified cardinality parameter of the network, $W$ its width, and $D$ its depth.

The construction will proceed in two steps:

1. **From Multi-Layered BNN into a BNN with one hidden layer.** In this step, we convert a deep BNN into an equivalent BNN with one hidden layer,
2. **From BNN with one hidden layer into an equivalent TT.** The resulting BNN with one hidden layer from the first step is then converted into an equivalent TT.

Next, we will provide details of how to perform each of these steps.

**Multi-Layered BNNs into BNNs with one hidden layer.** There are many ways to convert a multi-layered BNN into an equivalent BNN with one hidden layer. In the main paper, we propose a technique that collapses the network's depth progressively from the last layer up to the first layer. This technique has the advantage of better scaling with respect to the network's depth: Leveraging parallelization, this operation can be performed in $O(\log(D))$ using a similar parallel scan strategy presented in section D. A simpler method consists at settling for building a lookup table that maps the set of possible activation values of neurons in the first layer to the model's output. This lookup table can be built by enumerating all possible activation patterns of the first hidden layer and evaluating the model's output via a network's forward propagation operations. This operation runs in $O(2^{W_1} \cdot \texttt{poly}(D))$ time.

**BNNs with one hidden layer into TTs.** Given a BNN with one hidden layer whose width is equal to 1. The conversion to an equivalent TT can be achieved by remarking that a BNN with one hidden layer can be compiled into an equivalent DNF formula whose number of clauses is $O(2^{W_1})$.

Let $B$ be a BNN with one hidden layer over $n_{in}$ variables, and $W_1$ be the width of the first hidden layer. Denote:

- $\mathbf{n} = (n_1, \ldots, n_{W_i})$: the set of its hidden layer neuron activation predicates,
- $\mathcal{W} \in \{0,1\}^{\mathbb{D} \times 2^{\otimes W_1}}$: A Binary Tensor that maps the input to the activation pattern of the first hidden layer, i.e.

$$\forall(x_1, \ldots x_{n_{in}}, i_1, \ldots, i_{W_1}) \in \mathbb{D} \times [2]^{\otimes W_1} : \mathcal{W} = \begin{cases} 1 & \text{if } \bigwedge_{j=1}^{W_1} n_j(x_1, \ldots, x_{n_{in}}) \\ 0 & \text{otherwise} \end{cases} \quad (27)$$

- $\mathcal{T} \in \{0,1\}^{2^{\otimes n}}$: A binary tensor that arranges the lookup table (computed in the first step) that maps each activation pattern $\mathbf{n}$ to the model's output.

The function computed by the BNN $B$ can be written as:

$$f_B(x_1, \ldots, x_{n_{in}}) = \sum_{(i_1, \ldots, i_{W_1}) \in [2]^{\otimes W_1}} \mathcal{W}_{x_1, \ldots, x_{n_{in}}, i_1, \ldots i_{w_1}} \cdot \mathcal{T}_{i_1, \ldots, i_{W_1}} \quad (28)$$

Equation (28) expresses the function computed by the BNN $B$ as a contraction of two tensors: $\mathcal{W}$ and $\mathcal{T}$. The tensor $\mathcal{T}$ is nothing but a rearrangement of the lookup table constructed in the first step in a tensorized format. The construction of $\mathcal{T}$ requires $\mathcal{O}(2^{W_1})$ running time. On the other hand, even when the network's width is small, a naive construction of the tensor $\mathcal{W}$ (Equation (27)) may scale exponentially with the dimension of the input space. What we will show next is how, in bounded width and sparsity regime, this tensor admits a TT representation of size $\mathcal{O}(R^{W_1} \cdot \texttt{poly}(n_{in}, N))$.

**Construction of the tensor $\mathcal{W}$.** A first observation about the tensor $\mathcal{W}$ is that it encodes a CNF formula whose clauses correspond to activations of neurons in $\mathbf{n}$. On the other hand, in the context of BNNs, clauses are represented using reified cardinality constraints.

The construction procedure of a TT equivalent to the tensor $\mathcal{W}$ runs as follows:

1. For each neuron $n_j$, we construct its equivalent LDFA $\mathcal{A}^{n_j}$ using the procedure outlined as a warm-up at the beginning of this section. The size of the layer's state space of each of these LDFAs is bounded by $\mathcal{O}(R)$ ($R$ refers to the network's reified cardinality parameter).

2. Construct the resulting TT corresponding to the product of the LDFAs $\{\mathcal{A}^{n_j}\}_{j \in [W_1]}$ (subsection B.3), while keeping the rightmost core tensor with legs open, thereby ignore the final states reached by the TT.

Leveraging the Kronecker product operation over Finite State Automata [81] to perform product operations, the TT equivalent to the product of $W_1$ LDFAs (whose state space is bounded by a constant $R$) can be constructed in $\mathcal{O}(R^{W_1})$ time. Consequently, the entire construction produces a TT whose size is bounded by $\mathcal{O}(R^{W_1} \cdot n_{in} \cdot N)$. The running time complexity of the procedure is also bounded by $\mathcal{O}(R^{W_1} \cdot n_{in} \cdot N)$.

