# OpenReview forum: "SHAP Meets Tensor Networks: Provably Tractable Explanations with Parallelism"
_NeurIPS.cc/2025/Conference — NeurIPS 2025 poster_

### Official Review · Reviewer_KnUF · 2025-07-01

**Clarity:** 3
**Significance:** 3
**Originality:** 3
**Rating:** 5
**Confidence:** 4

**Summary:**

In this paper, the authors consider the problem of computing the SHAP
score for tensor networks. In particular, they prove that computing
the SHAP score for general tensor networks is #P-hard. Then they
consider the class of tensor trains, and show that for this class the
SHAP score can be computed not only in polynomial time, but also in
NC, which implies that it can be computed in time (log(n))^k for a
fixed constant k, assuming access to a polynomial number of parallel
processors. The authors use this result to strengthen the tractability
results for well-known ML models such as tree ensembles, decision
trees, linear RNNs, and linear models, concluding that for all these
models the SHAP score can be computed in NC (under tensor-train-based
distributions), which again implies that this computation is highly
parallelizable. Finally, the authors consider binarized neural
networks and show that the SHAP score can be computed in polynomial
time if the network width is bounded.

**Questions:**

Q1 In Proposition 3, the authors prove that the computation of the
SHAP score is in NC for the family of tensor trains. This implies that
such a computation is in NC^k for some k. The value of k is important
to understand the complexity of the parallel algorithm, so it should
be included in the paper.

Q2 What is the size of the circuits used in the proof of Proposition
3? These circuits are of size n^c for a fixed c, and again it is
important to know what the value of c is.

Q3 What is the width of the circuits used in the proof of Proposition
3? It is important to mention this in the paper as it tells us the
amount of parallelism needed to compute the SHAP score in
poly-logarithmic time.

**Ethical Concerns:**

["NO or VERY MINOR ethics concerns only"]

**Final Justification:**

I am satisfied with the authors’ answers to the reviewers’ comments. I still believe this paper makes a significant contribution, so I keep my acceptance score.

**Limitations:**

Yes

**Paper Formatting Concerns:**

No formatting issues

**Quality:**

4

**Strengths And Weaknesses:**

Strengths:

S1 The paper presents a fairly complete study of the complexity of
computing the SHAP score for useful ML models. In particular, it
proves that the SHAP score can be computed in polynomial time for the
class of tensor trains.

S2 The authors consider the possibility of parallelizing the
computation of the SHAP score, which is a useful way in practice to
reduce the computation time. In particular, they prove that the
computation of the SHAP score is highly parallelizable for the class
of tensor trains, from which they conclude that the same holds for
other well-known and widely used ML models.

Weaknesses:

W1 The paper is very dense, and it could benefit from some examples
and extra proof sketches. I understand the issue with the paper is the
lack of space, but still, the authors could have tried to save some
space to give a more thorough explanation of the polynomial-time
algorithm for tensor trains, and why it is highly parallelizable. This
intuition is something that the readers would appreciate.

---

> ### Author Rebuttal · Authors · 2025-07-31
>
> We thank the reviewer for their thorough, valuable, and constructive feedback and for acknowledging the significance of our work. See our detailed response below.
>
> **Improving the presentation and adding proof sketches**
>
> We thank the reviewer for acknowledging the theoretical contributions of our work. We agree that the slightly dense presentation in some parts of our work stems directly from the highly complex tensor-related technicalities involved in some of our proofs. Although our work already includes several illustrative and high-level examples in both the main text and appendix, we agree that adding more intuitive examples as well as proof sketches for central results, such as the efficient computation of Shapley-based explanations for the class of TTs and their parallelizability, would improve the clarity and accessibility of the paper. In response to this valuable suggestion, we plan to incorporate these additions in the final version, especially given the additional page allowance, as we agree they will strengthen the presentation of our results.
>
> **The width and depth of circuits for parallelized SHAP Computations**
>
> We thank the reviewer for highlighting these points. While some of these aspects are addressed in the appendix (e.g., lines 1214-1220), we fully agree that they are important and will incorporate a discussion of them in the main text in the final version.
>
> Formally, let us define the parameter $ n_{\text{in}}$ as the number of input features (the dimension of the input space) and let the terms $r_M$ and $r_P$ represent the maximal ranks of the TTs corresponding respectively to the model being explained and the probability distribution over the inputs. More precisely, the rank refers to the largest dimension among the tensor cores constituting each Tensor Train.
>
> Accordingly, we have that the *width* of the circuit for computing marginal SHAP values for TTs is bounded by:
>
>   $$ O\big(n_{\text{in}}^{5} \cdot r_M \cdot r_P\big) $$
>
> On the other hand, the *depth* of the circuit computing Marginal SHAP values for TTs is bounded by:
>
> $$ O\left(\log(n_{\text{in}}) \cdot \big[\log(n_{\text{in}}) + \log(r_M) + \log(r_P)\big]\right) $$
>
> Therefore, the problem of computing SHAP values for TTs belongs to the class $\text{NC}^{2}$. This complexity class includes many widely studied classic parallelizable problems such as *matrix multiplication* and *context-free grammar recognition*.
>
>  As such, our complexity results open up the door for leveraging both theoretical and practical insights developed for these problems in the context of SHAP computation. This observation is relevant for a very broad range of ML models, namely, those with expressivity no greater than that of TTs, as well as for expressive representations of distributions that can also be modeled by TTs. We thank the reviewer again for highlighting this point and agree that incorporating a discussion of it into the main text will strengthen the final version.

---

### Official Review · Reviewer_BJdq · 2025-07-01

**Clarity:** 3
**Significance:** 2
**Originality:** 3
**Rating:** 5
**Confidence:** 3

**Summary:**

The paper proves that Marginal SHAP values of a Tensor Trains (TT) model over a discrete domain can be efficiently computed with a parallel algorithm. It is also argued that TTs are expressive enough to represent Linear, Tree Ensembles, Linear-RNNs, so the results hints at a new algorithm that could rival existing ones (e.g. TreeSHAP) .

**Questions:**

## Data Distribution as a TT
Looking at the references provided at lines 267-269, it is not immediately clear to me that independent and empirical distributions can be modeled as a TT. It would help to provide a technical discussion (similar to Section D of the appendix) on how various distributions can be represented within this class of functions. This additional discussion should state clear assumptions about the data.

## Comparisons to Existing Algorithms
While I understand that the paper's contributions are meant to be mainly theoretical, it would be pertinent to highlight the practical
benefits of using Algorithm 1 on Tree Ensembles compared to TreeSHAP, LinearTreeSHAP, GPUTreeSHAP etc.

## Extended discussion on SHAP Tractability
There is ongoing work on designing new Neural Network architecture that have tractable Shapley Values [1]. While the current paper studies a different class of functions, the reference [1] and others within should be added in a **Related Work** section.

## Discretized Features
A fundamental assumption made by the Algorithm is that the feature $i$ has been discretized into $N_i$ values. This can be true for certain tree ensembles such as Histogram Gradient Boosted Trees which discretize features in order to efficiently compute statistics while varying the split threshold. However, this is not true for Random Forests for instance. Could the authors clarify the types of Tree Ensembles for which Algorithm 1 is relevant?

Moreover, the value $N_i$ can potentially be very large (for example $N_i=255$ by default if I use sklearn https://scikit-learn.org/stable/modules/generated/sklearn.ensemble.HistGradientBoostingClassifier.html). How would Marginal SHAP Tensor scale
if I varied $N_i$?

[1] Chen, Lu, et al. "HarsanyiNet: Computing Accurate Shapley Values in a Single Forward Propagation." International Conference on Machine Learning. PMLR, 2023.

**Ethical Concerns:**

["NO or VERY MINOR ethics concerns only"]

**Final Justification:**

Increased my score to "accept" given the author rebuttal, which adressed all my concerns.

**Limitations:**

The assumptions that features are discretized needs to be discussed in more depth. Also, it is not clear in which settings the data can be modeled with a TT.

**Quality:**

3

**Strengths And Weaknesses:**

# Strengths
- The work offers an original perspective on computing Marginal Shapley Values for a variety of models.
- The paper is technically sound and the theoretical section (2, 3, & 4) are well written.
- Theorem 1 is of both theoretical interest since it regroups a variety of ML models (.e.g. Tree Ensembles, Linear RNN), but also of potential practical interest because it suggests a parallel algorithm for computing Shapley values.

# Weaknesses
- It is assumed that the both the model and data distribution can be represented by a TT. While it proven in Appendix D that linear models
and tree ensembles can be represented with TT,  no such proofs are provided for realistic data distributions.
-  The paper does not compare Algorithm 1 (either empirically or theoretically)  to existing algorithms such as TreeSHAP, and its variants
LinearTreeSHAP [1], and GPUTreeSHAP [2].


## Minor Comments
- At line 191, i do not think that the expression $\mathbb{R}^{n_{in}\times 2^{\otimes n_{in}}}$ is correct. Indeed,
$2^{\otimes n_{in}}$ should be $2^{n_{in}}$ since you are counting the number of elements in the set $[2]^{\otimes n_{in}}$. Same thing at line 194.
- Algorithm 1 is not necessary and could be removed to increase the space. The method is already clear from the discussion at lines 197-200.
- $\mathcal{T}^M$ first introduced at Equation (2) is defined later at line 186. Introducing $\mathcal{T}^M, \mathcal{T}^P$ before Equation (2) in Section 4 would improve the flow.

[1] Bifet, Albert, Jesse Read, and Chao Xu. "Linear tree shap." Advances in Neural Information Processing Systems 35 (2022): 25818-25828.

[2] Mitchell, Rory, Eibe Frank, and Geoffrey Holmes. "GPUTreeShap: massively parallel exact calculation of SHAP scores for tree ensembles." PeerJ Computer Science 8 (2022): e880.

---

> ### Author Rebuttal · Authors · 2025-07-31
>
> We thank the reviewer for their valuable and constructive feedback and for acknowledging the significance of the theoretical contributions of our work. See our response below.
>
> **How can TTs model empirical and independent distributions?**
>
> We thank the reviewer for this valuable comment. The concern regarding the absence of a detailed proof for representing empirical and independent distributions using TTs is indeed valid. We would like to clarify our decision in this regard.
>
> Rather than reproducing a complete formal treatment, we chose to reference existing work [1], where this result is effectively established. Specifically, the cited work provides explicit constructions for converting both empirical and independent distributions into equivalent Hidden Markov Models (HMMs). Separately, it is well-known that HMMs are a particular instance of TTs [2]. This work is also referenced in our paper. Taken together, these two facts directly imply that empirical and independent distributions can be efficiently represented using TTs.
>
> Including a full proof of this fact, similar to those for other ML models in Appendix D, would have introduced a level of redundancy in our manuscript, given the availability of these results in prior work. For the sake of conciseness, we opted to cite the relevant literature instead.
>
> Moreover, we would like to point out that the expressive power of TTs extends well beyond these cases: TTs can model Markovian distributions [1] (and, by extension, $n$-gram models with reasonably small $n$), HMMs (as discussed above), uniform Matrix Product States (MPS), and Born Machines [3].
>
> We thank the reviewer for raising this concern. To address any potential ambiguity, we will make the connection between these classes of distributions and their representation using TTs in light of the relevant literature more explicit in the final version of the manuscript.
>
> **Comparison with other TreeSHAP variants**
>
> We thank the reviewer for requesting a comparative discussion. As summarized in the table below, our results improve upon the existing TreeSHAP variants reported in the literature across two dimensions: the complexity class of the SHAP computation algorithm and the class of probability distributions under which exact computation is feasible in reasonable time.
>
>
>    To clarify, traditional TreeSHAP algorithms (TreeSHAP, Linear TreeSHAP, Fast TreeSHAP), operate in polynomial time and generally fall within the $\mathbf{P}$ complexity class. The GPUTreeSHAP algorithm [6] improves computational efficiency by exploiting parallelization and fits within the $\mathbf{NC}$ complexity class. All these variants, however, focus primarily on empirical distributions. One of the key findings of our work, in contrast to existing TreeSHAP literature, is that it is possible to design fast, parallel algorithms for computing SHAP values for ensemble trees under significantly richer dependency structures.
>
> In summary, our method advances the state of the art by enabling efficient, parallelizable exact SHAP computations for ensemble trees under a substantially wider family of distributions, which is critical for capturing realistic data dependencies in practice. This represents a significant theoretical and practical improvement over the existing methods.
>
>
>  |  | **P** |   **NC\*** |
> |:----------:|:----------:|:----------:|
> | **Empirical**  | (Linear) TreeSHAP [4,5]  | GPUTreeSHAP [6]  |
> | **TTs (HMMs, Markovian, Empirical etc.)**  | Marzouk et al.'25\*\* [1]  | Ours  |
>
>
> \* $\scriptsize{\text{Note that the complexity class NC is included in P. Moreover, it is conjectured that this inclusion is strict.}}$
>
> \*\* $\scriptsize{\text{Work in [1] proved the tractability of computing Marginal SHAP of Ensemble Trees under the family of HMM distributions}}$
>
> **Tractable SHAP computation in Neural Networks**
>
> Thank you for this valuable feedback. We appreciate you pointing out this relevant line of work on neural network architectures with tractable Shapley value computation. In the final version of the paper, we will incorporate a more refined Related Work section that takes into account the reference you cited, as well as other related contributions in this direction, in order to better situate our approach within the broader literature.
>
> **SHAP for Ensemble Trees: Discrete vs Continuous Features**
>
> We thank the reviewer for raising this important point regarding the treatment of continuous features. While our theoretical analysis is developed under the assumption of a finite discrete feature space, our results also extend to cover SHAP computation of Decision Trees with axis-aligned split nodes of input continuous features.
>
> In axis-aligned decision trees, continuous features are typically handled through threshold-based splits of the form  $x_i \leq t$ or $x_i > t$. Each such condition can be viewed as a Boolean predicate, which may be explicitly introduced as a binary variable. By transforming continuous features into collections of such threshold predicates, the problem can be reformulated over discrete Boolean variables — consistent with the setting of our theoretical framework.
>
> Crucially, this transformation preserves the core structural properties that our complexity analysis relies upon. The number of threshold predicates is upper-bounded by the number of internal nodes in the ensemble, as each corresponds to a unique threshold condition. As a result, the total number of introduced variables remains polynomial in the size of the ensemble, and our **NC** parallel complexity result continues to hold under this extended setting.
>
> Regarding scalability with respect to $ N_i $, we note that the number of parallel processors required by our construction does not depend explicitly on this parameter. Instead, the parallelism level scales linearly with the number of input features $ n_{\text{in}} $ and the number of paths $L$ in the decision tree. More specifically, the total number of parallel processors required in our construction is upper bounded by $ O(n_{\text{in}} \cdot L) $ (see section D.2 in the appendix, lines 1305-1311).
>
> [1] On the Computational Tractability of the (Many) Shapley Values (Marzouk et al., AISTATS 2025)
>
> [2] Expressive Power of Tensor-Network Factorizations for Probabilistic Modeling (Glasser et al., NeurIPS 2019)
>
> [3] Tensor Networks for Probabilistic Sequence Modeling (Miller et al., AISTATS 2021)
>
> [4] From Local Explanations to Global Understanding with Explainable AI for Trees (Lundberg et al., Nature 2020)
>
> [5] Linear TreeSHAP (Yu et al., Neurips 2022)
>
> [6] GPUTreeShap: Massively Parallel ExactCalculation of SHAP Scores for Tree Ensembles (Mitchell et al., arXiv 2020)

---

> ### Comment · Reviewer_BJdq · 2025-08-04
> **Response to Rebuttal**
>
> I want to thank the authors for addressing my concerns. For that I will increase my score to "borderline accept".  I still want to follow up on some of the points that were brought up.
>
> Regarding the modeling of distributions as a TT. I went through the Appendix 5 of the reference [1] and I now understand how empirical distributions over binary features can be modeled as a HMM. I still think the paper could benefit from sharing some of the intuition without reproducing the proof. I think it is elegant to view an empirical distribution over 8 features as a random binary sequence 00110001 with stade transition probabilities ( e.g. P(001->0010) and P(001-> 0011)) computed by counting how many time a prefix occurs. Perhaps this intuition and the relationship with TT could be conveyed visually in an effective manner.
>
>
> On scalability w.r.t $N_i$, although the number of parallel processes for the method is independent of this quantity, I would expect the amount of work per process to depend on $N_i$. For instance the tensors $\mathcal{M}^{(i)}$ has a last dimension of size $N_i^2$. How would a  TT contraction algorithm avoid the complexity of summing across this dimension? I think my problem is that I do not understand the complexity of parallel TT contractions algorithms (memory complexity, number of processes, amount of work per processes). Adding an extended explanation of parallel TT contraction, as suggested by W1 of **Reviewer KnUF** could help.
>
> [1] On the Computational Tractability of the (Many) Shapley Values (Marzouk et al., AISTATS 20

---

> > ### Author Response · Authors · 2025-08-06
> >
> > We thank the reviewer for their valuable feedback and are glad that our rebuttal addressed the reviewer’s concerns and helped raise their score. Below, we provide a follow‑up addressing the reviewer’s additional points.
> >
> >
> > **Visual illustrations of how empirical distributions are converted into TTs**
> >
> >
> > Regarding the modeling of empirical distributions, we agree with the reviewer’s suggestion about incorporating additional visual illustrations to clarify the reduction strategies from empirical distributions to TTs. The approach hinted at by the reviewer (constructing a Probabilistic Prefix Tree Acceptor that implements a given empirical distribution and converting it into an equivalent TT) is indeed an intuitive way of conveying the underlying idea. We thank the reviewer for this great suggestion and will include such an illustration in the final version of the paper.
> >
> > **The scalability with respect to $N_i$**
> >
> > We thank the reviewer for raising this technical question regarding the scalability with respect to $N_i$ (i.e., the number of possible values the input feature $i$ can take). Before addressing the question, we would like to clarify a possible misunderstanding concerning the notions of “processes” and “amount of work per process” in the NC complexity class, as raised by the reviewer. This is indeed a very subtle and potentially confusing aspect of this complexity class, which we agree warrants clearer explanation.
> >
> > In the standard definition of NC, the model assumes *polynomially many processors*, each performing only a **constant amount of work** per step, over $O(\log^k n)$ parallel time steps, for some integer $k$. The complexity bound is therefore determined by the *total number of processors* and the *circuit depth*, not by the amount of computation per processor (which remains constant-time).
> >
> > In our result, the number of parallel processors required is independent of $N_i$. Since each processor is allowed to execute only constant-time operations per step, the amount of work per processor does not grow with $N_i$. The total computational work remains polynomial in the input size, consistent with the NC framework.
> >
> > Addressing the reviewer’s question of why the computational complexity does not depend explicitly on $N_i$: in the context of the SHAP computational problem (specifically the local version of SHAP, which is the focus of our paper), the parameter $N_i$ is involved only in selecting the matrix slice within $\mathcal{M}_i$ corresponding to the instance to explain. This selection can be performed in constant time.
> >
> > More intuitively, the tensor $\mathcal{M}_i$ (of order $3$) can be viewed as a collection of $N_i$ matrices. When an instance to explain is provided, the algorithm simply selects the relevant matrix for that instance from this collection. This indexing operation is performed in $O(1)$ time under the standard RAM model, and thus $N_i$ does not contribute to the running-time complexity of the method.
> >
> > However, $N_i$ does affect the *space complexity*, since the size of $\mathcal{M}_i$ grows linearly with $N_i$. Moreover, in the case of ensemble trees, there is an indirect dependency: Larger values of $N_i$ are more likely, in practice, to result in larger decision trees (when using classical ensemble tree learning algorithms), which in turn increases the computational complexity of our parallel algorithmic construction. Consequently, while $N_i$ does not enter the time complexity of our NC result explicitly, it may influence the computational cost indirectly through the model size in specific implementations such as decision tree ensembles.
> >
> > Following the suggestions of the reviewer and Reviewer KnUF, and given the importance of these complexity aspects, we will include a more detailed discussion of them in the main text of the final version.
> >
> >
> >
> > We thank the reviewer again for their great comments and questions, and would be glad to provide any additional clarification you may require.

---

> > > ### Comment · Reviewer_BJdq · 2025-08-06
> > > **Thank you**
> > >
> > > I wish to again thank the authors for their clarifications. I now understand that the runtime for *explaining a single instance* $x$ is independent of $N_i$ since the algorithm will select one of the $N_i$ slices of the $\mathcal{M}^{(i)}$ tensor based on the value of $x_i$. The authors have adressed all my points and so I am again increasing my score.

---

> > > > ### Author Response · Authors · 2025-08-07
> > > >
> > > > We thank the reviewer for their feedback and for the positive adjustment of the score. We are pleased that the rebuttal phase provided an opportunity to clarify the contribution of our work and to address the reviewer’s concerns constructively.

---

### Official Review · Reviewer_ycjs · 2025-07-02

**Clarity:** 2
**Significance:** 3
**Originality:** 3
**Rating:** 4
**Confidence:** 2

**Summary:**

This paper explores the complexity of computing (marginal) SHAP scores in tensor networks (TNs), and specific sub-classes, such as tensor trains (TTs), as well as binarized neural networks (BNNs). In Section 4, a general framework for computing SHAP scores in TNs is considered, where the SHAP score is expressed as a contraction of two tensors (Proposition 1), which allows to formulate Algorithm 1 to compute SHAP scores as a contraction of two tensors. Section 5 then discusses the complexity of this computation, where Proposition 2 shows that it is #P hard in general. Theorem 1 then establishes that if both $T^M$, and $T^P$ are TTs, then the SHAP tensor can also be expressed as a TT, which then yields that its complexity lies in NC (Proposition 3). As a consequence, decision trees, ensembles of decision trees, linear models and RNNs, can be expressed as TTs and are thus in NC (Theorem 2). Lastly, Section 6 discusses the computational complexity for BNNs, where it is shown (Theorem 3) that for any depth BNN computing SHAP scores is intractable, but narrow and particularly sparse narrow networks have lower complexity.

**Questions:**

- In Eq. 1 you define $\phi_i(M,x,P)$, which is then replaced in Section 4 by $\phi_i(T^M,x,T^P)$. Could you explain this new notation more carefully, e.g. what is $T^M,T^P$?
- What is the definition of $e^{N_1}_{x_1}$?
- What is the formal notion of sparsity used for BNNs?

**Ethical Concerns:**

["NO or VERY MINOR ethics concerns only"]

**Final Justification:**

While this paper is still very theoretical and hard to follow, I think that the contribution is interesting to the community. I appreciate the authors comments and changes, and I am leaning towards acceptance now. I won't increase my score further due to the limited practical implications, and my limited understanding of the technical details.

**Limitations:**

The authors discuss limitations of their work.

**Paper Formatting Concerns:**

I did not notice any formatting issues.

**Quality:**

3

**Strengths And Weaknesses:**

**Strengths**
- Exploring the complexity of SHAP scores for complex classes of networks is interesting
- The paper uses rigorous mathematical notation, and carefully distinguishes the complexity classes

**Weaknesses**
- The paper is very theoretical and abstract, and offers limited practical implications. E.g. the results shown in Theorem 2 for decision trees etc. were already known. Clearly, the class of TTs is more general, but there seems to be little practical implications from this generalization.
- For readers unfamiliar with TNs and TTs, the paper is generally very hard to follow. In my case, the illustrations did barely help to understand the general structure of TNs. Moreover, the paper claims that decision trees etc. can be expressed as TNs but does not explain further how this is possible. I guess that such simple examples could help the reader understand this abstract class of models.
- I appreciate the analysis of BNNs, but I would have expected a more practical approach here: What are the implications of this result in practice? What are the costs of enforcing theses constraints in BNNs, e.g. how much would this affect performance? Could you construct such BNNs and compute the SHAP scores, and evaluate the empirical complexity in terms of the parameters?

**Minor**
- typo line 272, "in in"

---

> ### Author Rebuttal · Authors · 2025-07-30
>
> We thank the reviewer for their valuable and constructive feedback and for acknowledging the significance of the theoretical contributions of our work. See our response below.
>
> **Implications of theoretical results for TTs and the novelty of their generalizability to other ML models**
>
> We appreciate the reviewer’s recognition of the novelty of our tractability result for SHAP within the class of TNs, which we believe is a significant contribution to the explainable AI community in general and to those focused on studying Tensor Networks in particular. The reviewer expressed concern about its practical relevance, suggesting that the results of Theorem 2 are already known. We respectfully clarify that this is not the case. Results of Theorem 2 advance the existing literature in two  ways, with clear practical implications:
>
> *First, improved expressive distribution modeling.* Our result shows that SHAP can be computed efficiently under distributions modeled by Tensor Trains (TTs), thereby relaxing prior distributional assumptions for exact Marginal SHAP computation. While recent work proves tractability for tree ensembles under Hidden Markov Models (HMMs) [1], our result extends this to distributions modeled by TTs. Notably, probabilistic TTs with non-negative weights coincide with HMMs, but general probabilistic TTs with arbitrary weights have been proven to be strictly more expressive [2]. This added expressiveness is important in practice, where incorporating realistic feature dependencies in the SHAP computation improves the quality of the delivered SHAP explanations [3–6].
>
> *Second, parallelizability across model classes:* Theorem 2 implies that SHAP can be efficiently parallelized for models with expressiveness up to that of TTs. One interesting corollary of our constructive proof of practical interest yields a generalized version of GPUTreeSHAP [7], extending its parallel GPU-enabled parallelized computation from empirical to more expressive distributions, such as HMMs and Born Machines [8].
>
> **More intuitive explanations of TNs and TTs and connections with tree ensembles**
>
> We thank the reviewer for this feedback. While the main paper and the appendix include illustrations and intuitions for TNs and TTs, we acknowledge that the highly technical nature of our treatment may pose challenges for readers less familiar with these models.
>
> To address this, we propose to improve the exposition in the main text by taking advantage of the available extra page to provide a more accessible introduction to TNs and TTs. Regarding our claim that tree ensembles can be transformed into equivalent TTs, we kindly invite the reviewer to read section D.2. in the appendix for formal details of how tree ensembles are NC-reducible to TTs. In the final version of the paper, we will make sure to include a simple visual example in the main paper illustrating how decision trees (and ensemble trees more generally) can be represented using TNs, to help ground the abstract formalism in concrete, familiar models.
>
> We believe these additions will significantly enhance the readability and pedagogical clarity of the paper.
>
> **The practicality of the analysis of BNNs**
>
> We thank the reviewer for their appreciation of our theoretical results on BNNs. We agree that these findings provide a novel fine-grained understanding of structural parameters in neural networks that influence the computation of Shapley values. While the impact of such structural parameters on the *learning* capabilities of neural networks is a central topic in deep learning theory (see, e.g., [9]), our work introduces a new theoretical perspective by linking them to interpretability/explainability considerations, and specifically, to the computation of Shapley-based explanations. While we agree that exploring the practical implications of our result is a natural next step, our primary goal in this work is to lay the theoretical groundwork for understanding which structural parameters of neural networks — particularly, *depth*, *width*, and *sparsity* — contribute to computational bottlenecks in explaining their predictions. In this sense, our contribution is theory-driven, aiming to characterize inherent complexity constraints that arise in BNNs from a structural standpoint.
>
> Investigating how these structural constraints can be incorporated into training pipelines without significantly degrading predictive performance, in order to obtain trained networks with provably efficient SHAP computations, is an important avenue for future work. We view our results as a foundation for such efforts.
>
> **Reply to questions**
>
> - In the preliminaries section, SHAP values are defined with respect to an arbitrary ML model $M$ and an arbitrary input distribution $P$. In Section 4, these are instantiated as $T^M$ and $T^P$, denoting the same objects (model and distribution) but implemented by TNs.
>
> - $e_{x_{i}}^{N_{i}}$ correspond to the one-hot vector of dimension $N_{i}$ where the element indexed by $x_{i}$ is equal to $1$. Intuitively, it refers to a single input feature fed to a TN model.
>
> - The notion of sparsity adopted in our work corresponds to the *reified cardinality parameter*, a structural measure introduced in [10] and briefly discussed in Section 3.3 (Lines 167–174). This parameter arises from an equivalent reified cardinality representation of BNNs and serves as a proxy for their sparsity. As shown in [10], constraining this parameter during training yields neural networks whose properties can be formally verified with remarkably efficient runtime performance.
>
> [1] On the Computational Tractability of the (Many) Shapley Values (Marzouk et al., AISTATS 2025)
>
> [2]   Expressive Power of Tensor-Network Factorizations for Probabilistic Modeling (Glasser et al., NeurIPS 2019)
>
> [3] Feature Synergy, Redundancy, and Independence in Global Model Explanations using SHAP Vector Decomposition (Ittner et al., arXiv 2021)
>
> [4] Multicollinearity Correction and Combined Feature Effect in Shapley Values (Basu et al., arXiv 2020)
>
> [5] Explaining Individual Predictions When Features Are Dependent: More Accurate Approximations to Shapley Values (Aas et al., arXiv 2021)
>
> [6] Explaining predictive models with mixed features using Shapley values and conditional inference trees (Redelmeier et al., arXiv 2020)
>
> [7] GPUTreeShap: Massively Parallel ExactCalculation of SHAP Scores for Tree Ensembles (Mitchell et al., arXiv 2020)
>
> [8] Tensor Networks for Probabilistic Sequence Modeling (Miller et al., AISTATS 2021)
>
> [9] Width is Less Important than Depth in ReLU Neural Networks (Vardi et al., COLT 2022)
>
> [10] Efficient Exact Verification of Binarized Neural Networks. (Jia et al., Neurips 2020)

---

> > ### Comment · Reviewer_ycjs · 2025-08-05
> > **Thank you**
> >
> > I thank the authors for addressing my concerns and helping me to further appreciate the contribution of this work. I apologize for missing Appendix D.2, which was very helpful. I decided to increase my score.

---

> > > ### Author Response · Authors · 2025-08-06
> > >
> > > We thank the reviewer for their valuable feedback and for the score increase. We appreciate your engagement and are glad that Appendix D.2 was helpful.

---

### Official Review · Reviewer_MUYn · 2025-07-07

**Clarity:** 2
**Significance:** 2
**Originality:** 2
**Rating:** 4
**Confidence:** 1

**Summary:**

Unfortunately, this paper falls outside my area of expertise. While I’m familiar with SHAP values, this paper focuses on theoretical results involving tensor networks and is filled with tensor operator formulations, a topic with which I have very limited experience.

Despite multiple readings, I find the paper extremely difficult to follow, which makes it challenging for me to provide a meaningful or fair review. I believe a reviewer with a stronger background in tensor networks or operations and quantum theory would be better suited for the task.

Thank you for your understanding.

**Questions:**

N/A

**Ethical Concerns:**

["NO or VERY MINOR ethics concerns only"]

**Final Justification:**

As I noted in my review, this paper falls outside my area of expertise. Please do not consider my rating.

**Quality:**

2

**Strengths And Weaknesses:**

N/A

---

> ### Author Rebuttal · Authors · 2025-07-30
>
> We appreciate the reviewer’s time, honesty, and feedback. We agree that several of our proofs involve highly technical and non-trivial tensor constructions, which may be challenging for readers unfamiliar with the field. That said, we believe that the figures, formalizations, and high-level intuitions included in the paper do help convey some of these concepts. Still, we fully agree that these aspects can be further refined in the final version, especially given the additional page allowance.
>
> While the implications of our results may be more immediately evident to researchers working on Tensor Networks, we also emphasize that our work has broad and significant relevance for the SHAP community. This includes extending configurations where SHAP can be computed exactly to popular ML models under substantially more general distributional assumptions using efficient parallelized algorithms.  Additionally, a novel fine-grained analysis of SHAP complexity for neural networks based on their inherent structural parameters was also presented.
>
> These contributions are clearly stated and presented in the introduction, conclusion, and throughout the main text, making them accessible even to SHAP researchers who may not engage deeply with every technical proof found in the appendix or the more rigorous sections. Nonetheless, we agree that refining the proof sketches, intuitive figures, and illustrations in the final version will enhance our paper’s accessibility.
>
>  We thank the reviewer once again for this valuable input.

---

### Decision · Program_Chairs · 2025-09-17

**Decision:**

Accept (poster)

**Comment:**

This paper studies the computational complexity of computing marginal Shapley values, and it focuses on the case of functions given by tensor networks and tensor trains in particular. The authors show that while computing Shapley values for tensor networks is #P hard in general, it can be computed in polynomial time (and in parallelized way) for tensor trains, which generalize decision trees, ensembles, linear models, and linear RNNS. The paper also shows that Shap is tractable for binarized networks if the width is bounded but remains hard with constant depth.

**Strengths**:
- Solid theoretical contribution
- Nice generalization of functions with tractable (and parallelizable) computation of marginal Shap.
- Potential implications for parallel implications for a broad class of functions (TTs).

**Weaknesses**
- Somewhat too abstract presentation and dense.
- Limited comparison to existing algorithms.
- The implications for better implementation are potential benefits, but the practical immediate benefits are limited.

**Discussion phase**:

This paper received 3 reviews (plus one, with minimal confidence). The discussion period was highly productive: the authors have clarified the position of their contribution in light of prior work, have promised to improve the presentation with clearer examples, visualizations, and proof sketches. The authors also made clarifications about their assumptions and their applicability.

**Conclusion**

The discussions above were well received by reviewers, who generally raised their scores and are supportive of the acceptance of this paper. I concur.